# On the linearity of large non-linear models: when and why the tangent kernel is constant

**Chaoyue Liu**
CSE
Ohio State University
liu.2656@osu.edu

**Libin Zhu**
CSE and HDSI
UC San Diego
l5zhu@ucsd.edu

**Mikhail Belkin**
HDSI
UC San Diego
mbelkin@ucsd.edu

## Abstract

The goal of this work is to shed light on the remarkable phenomenon of "transition to linearity" of certain neural networks as their width approaches infinity. We show that the "transition to linearity" of the model and, equivalently, constancy of the (neural) tangent kernel (NTK) result from the scaling properties of the norm of the Hessian matrix of the network as a function of the network width. We present a general framework for understanding the constancy of the tangent kernel via Hessian scaling applicable to the standard classes of neural networks. Our analysis provides a new perspective on the phenomenon of constant tangent kernel, which is different from the widely accepted "lazy training". Furthermore, we show that the "transition to linearity" is not a general property of wide neural networks and does not hold when the last layer of the network is non-linear. It is also not necessary for successful optimization by gradient descent.

## 1   Introduction [1]

As the width of certain non-linear neural networks increases, they become linear functions of their parameters. This remarkable property of large models was first identified in [10] where it was stated in terms of the constancy of the (neural) tangent kernel during the training process. More precisely, consider a neural network or, generally, a machine learning model $f(\mathbf{w}; \mathbf{x})$, which takes $\mathbf{x}$ as input and has $\mathbf{w}$ as its (trainable) parameters. Its tangent kernel $K_{(\mathbf{x}, \mathbf{z})}(\mathbf{w})$ is defined as follows:

$$K_{(\mathbf{x}, \mathbf{z})}(\mathbf{w}) := \nabla_{\mathbf{w}} f(\mathbf{w}; \mathbf{x})^T \nabla_{\mathbf{w}} f(\mathbf{w}; \mathbf{z}), \quad \text{for fixed inputs } \mathbf{x}, \mathbf{z} \in \mathbb{R}^d. \quad (1)$$

The key finding of [10] was the fact that for some wide neural networks the kernel $K_{(\mathbf{x}, \mathbf{z})}(\mathbf{w})$ is a constant function of the weight $\mathbf{w}$ during training. While in the literature, including [10], this phenomenon is described in terms of the (linear) training dynamics, it is important to note that the tangent kernel is associated to the model itself. As such, it does not depend on the optimization algorithm or the choice of a loss function.

The goal of this work is to clarify a number of issues related to the constancy of the tangent kernel, to provide specific conditions when the kernel is constant, i.e., when non-linear models in the limit, as their width approach infinity, become linear, and also to explicate the regimes when they do not. One important conclusion of our analysis is that the "transition to linearity" phenomenon discussed in this work (equivalent to constancy of tangent kernel) cannot be explained by "lazy training" [5] (often described as small change of parameters from the initialization point), which is widely held to be the reason for constancy of the tangent kernel, e.g., [17, 2, 9]. The "transition to linearity" is neither due to a choice of a scaling of the model (see Appendix A), nor is a universal property of large models

including infinitely wide neural networks. In particular, the models shown to "transition to linearity" in this paper become linear in a Euclidean ball of an arbitrary fixed radius, not just in the vicinity of the initialization point.

Our first observation[2] is that a function $f(\mathbf{w}, \mathbf{x})$ has a constant tangent kernel *if and only if* it is linear in $\mathbf{w}$, that is for some function $\phi$

$$f(\mathbf{w}, \mathbf{x}) = \mathbf{w}^T \phi(\mathbf{x}).$$

Thus the constancy of the tangent kernel is directly linked to the linearity of the underlying model.

So what is the underlying reason that some large models "transition to linearity" as a function of the parameters and when do we expect it to be the case? As known from the mathematical analysis, the deviation from the linearity is controlled by the second derivative, which is represented, for a multivariate function $f$, by the *Hessian matrix $H$*. If its spectral norm $\|H\|$ is small compared to the gradient $\nabla_{\mathbf{w}} f$ in a ball of a certain radius, the function $f$ will be close to linear and will have near-constant tangent kernel in that ball. Crucially, the spectral norm $\|H\|$ depends not just on the magnitude of its entries, but also on the structure of the matrix $H$. This simple idea underlies the analysis in this paper. Note that throughout this paper we consider the Hessian of the model $f$, not of any related loss function.

**Constant tangent kernel for neural networks with linear output layer.** In what follows we analyze certain classes of standard neural networks, including those which have been found to have constant tangent kernel in [10, 13, 6] and other works. We show that while the gradient norm $\|\nabla_{\mathbf{w}} f\|$ is (omitting log factors) of the order $\Theta(1)$, the spectral norm of the Hessian matrix $\|H\|$ scales with the width $m$ as $1/\sqrt{m}$. In the infinite width limit this implies "transition to the linearity" of the model in a ball of an arbitrary fixed radius. A consequence of this is the constancy of the tangent kernel, providing a different perspective on the results in [10] and the follow-up works.

We proceed to expose the underlying reason why the Hessian matrix scales differently from the gradient and delimit the regimes where this phenomenon exists. As we show, the scaling of the Hessian norm relates to two structural properties of (certain) neural networks: (a) *linear output layer*, (b) *dependence of non-linear activations on a single linear "preactivation" neuron*[3].

In particular, for shallow networks these two properties result in the sparsity of the Hessian matrix. It is easy to see that the spectral norm of a sparse matrix is of the same order as its entries, $O(1/\sqrt{m})$. The case of deep networks is more subtle as their Hessian matrices are not generally sparse. In this work we develop a general recursive condition, showing that inserting an additional wide layer into a deep neural network with a linear output layer does not affect the scaling of the Hessian in terms of $m$. We show how these criteria follow the structural properties (a) and (b) for several classes of standard deep networks, including combinations of fully connected, convolutional and residual layers.

**Non-constancy of tangent kernels.** We proceed to demonstrate, both theoretically (Section 4) and experimentally (Section 6), that the constancy of tangent kernel is not a general property of large models, including wide networks, even in the "lazy" training regime. In particular, if the output layer of a network is nonlinear, e.g., if there is a non-linear activation on the output, the Hessian norm does not tend to zero as $m \to \infty$, and constancy of tangent kernel will not hold in any fixed neighborhood and along the optimization path, although each individual parameter may undergo only a small change. This demonstrates that the constancy of the tangent kernel relies on specific structural properties of the models. Similarly, we show that inserting a narrow "bottleneck" layer, even if it is linear, will generally result in the loss of near-linearity, as the Hessian norm becomes large compared to the gradient $\nabla_{\mathbf{w}} f$ of the model.

Importantly, as we discuss in Section 5, non-constancy of the tangent kernel does not preclude efficient optimization. We construct examples of wide networks which can be provably optimized by gradient descent, yet with tangent kernel provably far from constant along the optimization path and with Hessian norm $\Omega(1)$, same is the gradient.

We finish this section with some important comments.

**"Small" weight change from initialization does not explain constant NTK.** It is sometimes stated in recent literature [17, 2, 9] that constancy of tangent kernel is a consequence of the small change in weight vector of wide networks during training, which is termed "lazy training" in [5]. It is important to note that the notion of "small" change depends crucially on the measurement. Let $\mathbf{w}_0$ and $\mathbf{w}^*$ be the weight vectors at initialization and at convergence respectively. For example, consider a one hidden layer network of width $m$. Each *component* of the weight vector is updated by $O(1/\sqrt{m})$ under gradient descent, as shown in [10], and hence for wide networks $\|\mathbf{w}^* - \mathbf{w}_0\|_\infty = O(1/\sqrt{m})$, a quantity that vanishes with the increasing width. In contrast, the change of the *Euclidean norm* is not small in training, $\|\mathbf{w}^* - \mathbf{w}_0\|^2 = \sum_{i=1}^m (w_i^* - w_{0,i})^2 = O(1)$. Thus convergence happens within a Euclidean ball with radius independent of the network width.

In fact, the Euclidean norm of the change of the weight vector cannot be small for Lipschitz continuous models, even in the limit of infinite parameters. This is because

$$\|\mathbf{w}^* - \mathbf{w}_0\| \geq |f(\mathbf{w}_0; \mathbf{x}) - y| / \sup_{\mathbf{w}} \|\nabla_{\mathbf{w}} f(\mathbf{w})\| \tag{2}$$

where is $y$ is the label at $\mathbf{x}$. Since $|f(\mathbf{w}_0; \mathbf{x}) - y| = \Theta(1)$ and $\|\nabla_{\mathbf{w}} f\| = O(1)$, we see that $\|\mathbf{w}^* - \mathbf{w}_0\| = \Omega(1)$, no matter how many parameters the model $f$ has.

We note that (approximate) linearity of a model in a certain region (and hence constancy of the tangent kernel) is closely related to the second-order term of the Taylor expansion $(\mathbf{w} - \mathbf{w}_0)^T H (\mathbf{w} - \mathbf{w}_0)$. This term is controlled by the *Euclidean distance* from the initialization $\|\mathbf{w} - \mathbf{w}_0\|$ (and the spectral norm of the Hessian), instead of the $\infty$-norm $\|\mathbf{w} - \mathbf{w}_0\|_\infty$.

**Rescaling model outputs**. We note that as discussed in [5], rescaling the model outputs could also lead to constant tangent kernel. However, as we discuss in Appendix A, this does not explain the phenomenon of constant tangent kernel observed in [10].

In contrast to these interpretations, we show that certain large networks have near constant tangent kernel in a ball of fixed radius due to vanishing Hessian norm, as their widths approach infinity. Indeed, that is the case for networks analyzed in the NTK literature [10, 13, 6, 7].

**Linearity vs optimization.** We note that the transition to linearity discussed in this paper implies convergence of gradient descent assuming that the tangent kernel is non-degenerate at initialization. However, it is important to emphasize that the linearity or near-linearity is not a necessary condition for convergence. Instead, convergence is implied by uniform conditioning of the tangent kernel in a neighborhood of a certain radius, while the linearity is controlled by the norm of the Hessian. These are conceptually and practically different phenomena as we show with wide shallow network with a non-linear output layer in Section 5. See also [14] for an in-depth discussion of optimization.

## 2 Tangent kernel and the Hessian

**Notation.** We use bold lowercase letters, e.g., $\mathbf{v}$, to denote vectors, capital letters, e.g., $W$, to denote matrices, and bold capital letters, e.g., $\mathbf{W}$, to denote matrix tuples or higher order tensors. We denote the set $\{1, 2, \cdots, n\}$ as $[n]$. For vectors, we use $\|\cdot\|$ to denote the Euclidean norm and $\|\cdot\|_\infty$ for the $\infty$-norm (i.e., the max norm). For matrices, we use $\|\cdot\|$ to denote the spectral norm (i.e., 2-norm) and $\|\cdot\|_F$ to denote the Frobenius norm. We use $\nabla_{\mathbf{w}} f$ and $\nabla_{\mathbf{x}} f$ to represent the first derivatives of $f(\mathbf{w}; \mathbf{x})$ with respect to its arguments $\mathbf{w}$ and $\mathbf{x}$ respectively. In addition, we use tilde, e.g., $\tilde{O}(\cdot)$, to hide logarithmic terms in Big-O notation. $\square$

Consider a machine learning model, e.g., a neural network, $f(\mathbf{w}; \mathbf{x})$, which takes $\mathbf{x} \in \mathbb{R}^d$ as input and has $\mathbf{w} \in \mathbb{R}^p$ as the trainable parameters. Throughout this paper, we assume $f$ is twice differentiable with respect to the parameters $\mathbf{w}$. To simplify the analysis, we further assume the output of the model $f$ is a scalar. Given a set of points $\{\mathbf{x}_i\}_{i=1}^n$, where each $\mathbf{x}_i \in \mathbb{R}^d$, one can build a $n \times n$ tangent kernel matrix $K(\mathbf{w})$, where each entry $K_{ij}(\mathbf{w}) = K_{(\mathbf{x}_i, \mathbf{x}_j)}(\mathbf{w})$.

As discovered in [10] and analyzed in the consequent works [13, 6] the tangent kernel is constant for certain infinitely wide networks during training by gradient descent methods. First, we observe that the constancy of the tangent kernel is equivalent to linearity of the model. While the mathematical result is not new (see [8, 16]), we have not seen this stated in the machine learning literature (the proof can be found in Appendix D).

**Proposition 2.1** (Constant tangent kernel = Linear model). *The tangent kernel of a differentiable function $f(\mathbf{w}; \mathbf{x})$ is constant if and only if $f(\mathbf{w}; \mathbf{x})$ is linear in $\mathbf{w}$.*

The following proposition shows that small Hessian norm is a sufficient condition for near-constant tangent kernel. The proof can be found in Appendix E.

**Proposition 2.2** (Small Hessian norm $\Rightarrow$ Small change of tangent kernel). *Given a point $\mathbf{w}_0 \in \mathbb{R}^p$ and a ball $B(\mathbf{w}_0, R) := \{\mathbf{w} \in \mathbb{R}^p : \|\mathbf{w} - \mathbf{w}_0\| \leq R\}$ with fixed radius $R > 0$, if the Hessian matrix satisfies $\|H(\mathbf{w})\| < \epsilon$, where $\epsilon > 0$, for all $\mathbf{w} \in B(\mathbf{w}_0, R)$, then the tangent kernel $K(\mathbf{w})$ of the model, as a function of $\mathbf{w}$, satisfies*

$$|K_{(\mathbf{x},\mathbf{z})}(\mathbf{w}) - K_{(\mathbf{x},\mathbf{z})}(\mathbf{w}_0)| = O(\epsilon R), \quad \forall \mathbf{w} \in B(\mathbf{w}_0, R), \; \forall \mathbf{x}, \mathbf{z} \in \mathbb{R}^d. \tag{3}$$

As we shall see in Section 3, all neural networks that are proven in [10, 6, 7] to have (near) constant tangent kernel during training, have small (zero, in the limit of $m \to \infty$) spectral norms of the corresponding Hessian matrices.

# 3 Linearity of large non-linear networks with linear output layers

In this section, we analyze the class of neural networks with linear output layers. We show that the spectral norm of the Hessian matrix becomes small, when the width of each hidden layer increases. In the limit of infinite width, these spectral norms vanish and the models become linear, with constant tangent kernels.

## 3.1 Shallow Neural Networks

As a warm-up for the more complex setting of deep networks, we start by considering the simple case of a shallow fully-connected neural network with a fixed output layer, defined as follows:

$$f(\mathbf{w}; x) = \frac{1}{\sqrt{m}} \sum_{i=1}^{m} v_i \alpha_i(x), \text{with } \alpha_i(x) = \sigma(w_i x), \; x \in \mathbb{R}. \tag{4}$$

Here $m$ is the number of neurons in the hidden layer, $\mathbf{v} = (v_1, \cdots, v_m)$ is the vector of output layer weights, $\mathbf{w} = (w_1, \cdots, w_m) \in \mathbb{R}^m$ is the weights in the hidden layer. We assume that the activation function $\sigma(\cdot)$ is $\beta_\sigma$-smooth. We initialize at random, $w_i \sim \mathcal{N}(0, 1)$ and $v_i \in \{-1, 1\}$. We treat $\mathbf{v}$ as fixed parameters and $\mathbf{w}$ as trainable parameters. For the purpose of illustration, we assume the input $x$ is of dimension 1, and the multi-dimensional analysis is similar.

**Remark 3.1.** This definition of a shallow neural network (i.e., with the presence of a factor $1/\sqrt{m}$ and $v_i$ and $w_i$ of order $O(1)$) is consistent with the NTK parameterization used to show constancy of tangent kernel in [10, 13].

**Structural properties.** We point out that the neural network $f$ has the following two structural properties, which will be key in our discussion:

(a) *linear output layer:* The output $f$ is a linear combination of the hidden units $\alpha_i(x)$.

(b) *sparse dependence of non-linearity:* each non-linear activation $\alpha_i$ depends on a single preactivation $\tilde{\alpha}_i = w_i x$.

**Hessian matrix.** We observe that (as a consequence of the structural properties above) the Hessian matrix $H$ of the neural network $f$ is sparse, specifically, diagonal:

$$H_{ij} = \partial^2 f / \partial w_i \partial w_j = \frac{1}{\sqrt{m}} v_i \sigma''(w_i x) x^2 \, \mathbb{1}_{\{i=j\}}.$$

Consequently, if the input $x$ is bounded, say $|x| \leq C$, the spectral norm of the Hessian $H$ is

$$\|H\| = \max_{i \in [m]} |H_{ii}| = \frac{x^2}{\sqrt{m}} \max_{i \in [m]} |v_i \sigma''(w_i x)| \leq \frac{1}{\sqrt{m}} \beta_\sigma C^2 = O\left(\frac{1}{\sqrt{m}}\right). \tag{5}$$

In the limit of $m \to \infty$, the spectral norm $\|H\|$ converges to 0.

**Tangent kernel and gradient.** On the other hand, the magnitude of the norm of the tangent kernel of $f$ is of order $\Theta(1)$ in terms of $m$. Specifically, for each diagonal entry we have

$$K_{(x,x)}(\mathbf{w}) = \|\nabla_{\mathbf{w}} f(\mathbf{w}; x)\|^2 = \frac{1}{m} \sum_{i=1}^{m} x^2 (\sigma'(w_i x))^2 = \Theta(1). \tag{6}$$

In the limit of $m \to \infty$, $K_{(x,x)}(\mathbf{w}) = x^2 \mathbb{E}[(\sigma'(wx))^2]$. Hence the trace of tangent kernel is also $\Theta(1)$. Since the tangent kernel is a positive definite matrix of size independent of $m$, the norm is of the same order as the trace.

Therefore, from Eq. (5) and Eq. (6) we observe that the tangent kernel scales as $\Theta(1)$ while the norm of the Hessian scales as $O(1/\sqrt{m})$ with the size of the neural network $f$. Furthermore, as $m \to \infty$, the norm of the Hessian converges to zero and, by Proposition 2.2, the tangent kernel becomes constant.

To provide a simple example, we assumed that the weights of the last layer $\mathbf{v}$ are fixed. It is not necessary. For a shallow network with a trainable output layer, the Hessian is a block matrix with three sparse blocks and a block of zeros. An analysis parallel to the one above still applies. This case is subsumed in the more general discussion of deep networks which follows.

## 3.2 Deep neural networks with linear output layer

We will now show that the spectral norm of the Hessian matrix for a deep neural network with a linear output layer is also of the order $\tilde{O}(1/\sqrt{m})$, where $m$ is the width of the network.

We consider a very general form of a deep neural network $f$ with a linear output layer:

$$
\begin{aligned}
\alpha^{(0)} &= \mathbf{x}, \\
\alpha^{(l)} &= \phi_l(\mathbf{w}^{(l)}; \alpha^{(l-1)}), \ \forall l = 1, 2, \cdots, L, \\
f &= \frac{1}{\sqrt{m}} \mathbf{v}^T \alpha^{(L)},
\end{aligned}
\tag{7}
$$

where each vector-valued function $\phi_l(\mathbf{w}^{(l)}; \cdot) : \mathbb{R}^{m_{l-1}} \to \mathbb{R}^{m_l}$, with parameters $\mathbf{w}^{(l)} \in \mathbb{R}^{p_l}$, is considered as a layer of the network. This definition includes the fully connected, convolutional (CNN) and residual (ResNet) neural networks as special examples. The factor $1/\sqrt{m}$ in the output layer is required by the NTK parameterization in order that the output $f$ is of order $\Theta(1)$. This is also consistent with the setting in [10].

**Initialization and parameterization.** In this paper, we consider the NTK initialization/ parameterization [10], under which the constancy of the tangent kernel had been initially observed. Specifically, the parameters, (a.k.a., weights), $\mathbf{W} := \{\mathbf{w}^{(1)}, \mathbf{w}^{(2)}, \cdots, \mathbf{w}^{(L)}, \mathbf{w}^{(L+1)} := \mathbf{v}\}$ are drawn i.i.d. from a standard Gaussian, i.e., $w_i^{(l)} \sim \mathcal{N}(0, 1)$, at initialization, denoted as $\mathbf{W}_0$. Different parameterizations (e.g., LeCun initialization: $w_i^{(l)} \sim \mathcal{N}(0, 1/m)$) rescale the tangent kernel and the Hessian by the same factor, and thus do not change our conclusions (see Appendix B).

**Assumptions.** We assume the hidden layer width $m_l = m$ for all $l \in [L]$, the output is a scalar, and the number of parameters in each layer $p_l \geq m$. We assume that (vector-valued) layer functions $\phi_l(\mathbf{w}; \alpha), l \in [L]$, are Lipschitz continuous and twice differentiable w.r.t. input $\alpha$ and parameters $\mathbf{w}$.

For each layer function $\phi_l(\mathbf{w}; \alpha)$, there exist three order 3 tensors: $\frac{\partial^2 \phi_l}{\partial \mathbf{w}^2} \in \mathbb{R}^{p_l \times p_l \times m}$, $\frac{\partial^2 \phi_l}{\partial \alpha^2} \in \mathbb{R}^{m \times m \times m}$ and $\frac{\partial^2 \phi_l}{\partial \alpha \partial \mathbf{w}} \in \mathbb{R}^{m \times p_l \times m}$. Given an order 3 tensor $\mathbf{T} \in \mathbb{R}^{d_1 \times d_2 \times d_3}$, with components $T_{ijk}, i \in [d_1], j \in [d_2], k \in [d_3]$, define its $(2, 2, 1)$-norm as

$$
\|\mathbf{T}\|_{2,2,1} := \sup_{\|\mathbf{x}\| = \|\mathbf{z}\| = 1} \sum_{k=1}^{d_3} \left| \sum_{i=1}^{d_1} \sum_{j=1}^{d_2} T_{ijk} x_i z_j \right|, \quad \text{where } \mathbf{x} \in \mathbb{R}^{d_1}, \mathbf{z} \in \mathbb{R}^{d_2}.
\tag{8}
$$

In the following theorem, we identify structural conditions that are sufficient to induce a small Hessian spectral norm. See proof in Appendix F.

**Theorem 3.1.** *Consider a neural network of the form in Eq.(7). Suppose, the following conditions hold, for all layers $l \in [L]$ and for all parameters $\mathbf{W} \in \mathcal{S} \subset \mathbb{R}^p$:*

*(a) Layer-wise quantities $\|\frac{\partial^2 \phi_l}{\partial \mathbf{w}^2}\|_{2,2,1}$, $\|\frac{\partial^2 \phi_l}{\partial \alpha^2}\|_{2,2,1}$ and $\|\frac{\partial^2 \phi_l}{\partial \alpha \partial \mathbf{w}}\|_{2,2,1}$ are of order $\tilde{O}(1)$;*

*(b) $\|\nabla_{\alpha^{(l)}} f\|_\infty$ is of order $\tilde{O}(1/\sqrt{m})$.*

*Then, the Hessian matrix $H$ satisfies $\|H\| = \tilde{O}(\frac{1}{\sqrt{m}})$ in the domain $\mathcal{S}$.*

**Remark 3.2.** The intuition for condition (b) is that there should be no dominant neurons within each layer during back propagation.

When Theorem 3.1 applies, in the limit of infinite network width, i.e. $m \to \infty$, the Hessian spectral norm $\|H\| \to 0$. By Proposition 2.2, such neural networks have constant tangent kernels within $\mathcal{S}$, if $\mathcal{S}$ has a finite radius.

In the following, we show that fully connected neural networks satisfy the structural conditions in Theorem 3.1, hence, its Hessian spectral norm scales as $\tilde{O}(1/\sqrt{m})$, in a region with finite radius. In Appendix H, we show these conditions are also satisfied by deep CNNs and ResNets.

### 3.3 Fully connected neural networks

A hidden layer of a fully connected neural network is given by

$$\alpha^{(l)} = \sigma(\tilde{\alpha}^{(l)}), \;\; \tilde{\alpha}^{(l)} = \frac{1}{\sqrt{m}} W^{(l)} \alpha^{(l-1)}, \text{ for } l \in [L], \tag{9}$$

where $\sigma(\cdot)$ is a $L_\sigma$-Lipschitz continuous, $\beta_\sigma$-smooth activation function. The layer parameters $W^{(l)}$ are reshaped into an $m \times m$ matrix. The Euclidean norm of $\mathbf{W}$ becomes: $\|\mathbf{W}\| = (\sum_{l=1}^{L} \|W^{(l)}\|_F^2)^{1/2}$.

**Theorem 3.2.** *Consider a neural network $f(\mathbf{W}; \mathbf{x})$, Eq.(7), with fully connected layers Eq.(9). Given any fixed $R > 0$, and any $\mathbf{W} \in B(\mathbf{W}_0, R) := \{\mathbf{W} : \|\mathbf{W} - \mathbf{W}_0\| \leq R\}$, the Hessian spectral norm satisfies the following, with high probability:*

$$\|H(\mathbf{W})\| = \tilde{O}\left(1/\sqrt{m}\right). \tag{10}$$

See proof of the theorem in Appendix G.

**Sparse dependence of the non-linear activation functions.** We observe that the sparse dependence of the non-linear activation functions $\sigma(\cdot)$ plays a key role in making the fully connected neural network satisfy the condition (b). Specifically, by Eq.(9), we have

$$\nabla_{\alpha^{(l-1)}} f = \frac{1}{\sqrt{m}} (W^{(l)})^T \Sigma'(\tilde{\alpha}^{(l)}) \nabla_{\alpha^{(l)}} f,$$

where $\Sigma'(\tilde{\alpha}^{(l)})$ is a diagonal matrix, with entry $\Sigma'_{ii} = \sigma'(\tilde{\alpha}^{(l)}_i)$. The sparsity of the matrix $\Sigma'$ stems from the fact that the non-linearity $\sigma(\cdot)$ is applied to each neuron individually, i.e., each activated neuron $\alpha^{(l)}_i$ depends on only one preactivated neuron $\tilde{\alpha}^{(l)}_i$. This sparsity allows the condition on $\|\nabla_{\alpha^{(l)}} f\|_\infty$ of $l$-th layer transits to the vector $\|\Sigma'(\tilde{\alpha}^{(l)}) \nabla_{\alpha^{(l)}} f\|_\infty$, and ultimately transits to $\|\nabla_{\alpha^{(l-1)}} f\|_\infty$ of $(l-1)$-th layer. Recursively applying this transition, the condition (b) holds for every layer.

In the limit of $m \to \infty$, the spectral norm of the Hessian $\|H(\mathbf{W})\|$ converges to 0, for all $\mathbf{W} \in B(\mathbf{W}_0, R)$, while the gradient norm $\|\nabla_{\mathbf{w}} f(\mathbf{w})\|$ is generically of the order $\Theta(1)$. By Proposition 2.2, this immediately implies constancy of tangent kernel and linearity of the model, in the ball $B(\mathbf{W}_0, R)$.

**Remark 3.3.** By the optimization theory built in our work [14], a finite radius $R$ is enough to include the gradient descent solution, for the square loss. Hence, for very wide networks, the tangent kernel is constant during gradient descent training.

**Architecture with mixed layer types.** Neural networks used in practice can be a mixture of differently layer types, such as having fully connected layers after a series of convolutional layers. Since the conditions in Theorem 3.1 and our analysis are layer-wise, they also apply to such neural networks. For a detailed discussion, see Appendix H.3.

## 4 Constant tangent kernel is not a general property of wide networks

In this section, we show that a class of infinitely wide neural networks with *non-linear* output, do not generally have constant tangent kernels. It also demonstrates that a linear output layer is a necessary condition for "transition to linearity".

We consider the neural network $\tilde{f}$:

$$\tilde{f}(\mathbf{w}; \mathbf{x}) := \phi(f(\mathbf{w}; \mathbf{x})). \tag{11}$$

where $f(\mathbf{w}; \mathbf{x})$ is a sufficiently wide neural network with linear output layer considered in Section 3, and $\phi(\cdot)$ is a non-linear twice-differentiable activation function. The only difference between $f$ and $\tilde{f}$ is that $\tilde{f}$ has a non-linear output layer. As we shall see, this difference leads to a non-constant tangent kernel during training, as well as a different scaling behavior of the Hessian spectral norm.

**Tangent kernel of $\tilde{f}$.** The gradient of $\tilde{f}$ is given by $\nabla_{\mathbf{w}}\tilde{f}(\mathbf{w}; \mathbf{x}) = \phi'(f(\mathbf{w}; \mathbf{x}))\nabla_{\mathbf{w}}f(\mathbf{w}; \mathbf{x})$. Hence, each diagonal entry of the tangent kernel of $\tilde{f}$ is

$$\tilde{K}_{(\mathbf{x},\mathbf{x})}(\mathbf{w}) = \|\nabla_{\mathbf{w}}\tilde{f}(\mathbf{w}; \mathbf{x})\|^2 = \phi'^2(f(\mathbf{w}; \mathbf{x}))K_{(\mathbf{x},\mathbf{x})}(\mathbf{w}), \tag{12}$$

where $K_{(\cdot,\cdot)}(\mathbf{w})$ is the tangent kernel of $f$. By Eq.(6) we have $\tilde{K}_{(\mathbf{x},\mathbf{x})}(\mathbf{w}) = \Theta(1)$, which is of the same order as $K_{(\mathbf{x},\mathbf{x})}(\mathbf{w})$.

Yet, unlike $K_{(\cdot,\cdot)}(\mathbf{w})$, the kernel $\tilde{K}_{(\cdot,\cdot)}(\mathbf{w})$ changes significantly during training, even as $m \to \infty$ (with a change of the order of $\Theta(1)$). To prove that, it is enough to verify that at least one entry of $\tilde{K}_{(\mathbf{x},\mathbf{x})}(\mathbf{w})$ has a change of $\Theta(1)$, for an arbitrary $\mathbf{x}$. Consider a diagonal entry. For any $\mathbf{w}$, we have

$$\left|\tilde{K}_{(\mathbf{x},\mathbf{x})}(\mathbf{w}) - \tilde{K}_{(\mathbf{x},\mathbf{x})}(\mathbf{w}_0)\right| = \left|\phi'^2(f(\mathbf{w}; \mathbf{x}))K_{(\mathbf{x},\mathbf{x})}(\mathbf{w}) - \phi'^2(f(\mathbf{w}_0; \mathbf{x}))K_{(\mathbf{x},\mathbf{x})}(\mathbf{w}_0)\right|$$

$$\geq \underbrace{\left|\phi'^2(f(\mathbf{w}; \mathbf{x})) - \phi'^2(f(\mathbf{w}_0; \mathbf{x}))\right| \cdot K_{(\mathbf{x},\mathbf{x})}(\mathbf{w}_0)}_{A} - \underbrace{\phi'^2(f(\mathbf{w}; \mathbf{x})) \cdot \left|K_{(\mathbf{x},\mathbf{x})}(\mathbf{w}) - K_{(\mathbf{x},\mathbf{x})}(\mathbf{w}_0)\right|}_{B}.$$

We note that the term $B$ vanishes as $m \to \infty$ due to the constancy of the tangent kernel of $f$. However the term $A$ is generally of the order $\Theta(1)$, when $\phi$ is non-linear[4]. To see that consider any solution $\mathbf{w}^*$ such that $f(\mathbf{w}^*; \mathbf{x}) = y$ (which exists for over-parameterized networks). Since $f(\mathbf{w}_0; \mathbf{x})$ is generally not equal to $y$, we obtain the result.

**"Lazy training" fails to explain constancy of NTK.** From the above analysis, we can see that, even with the same parameter settings as $f$ (i.e., same initial parameters and same parameter change), network $\tilde{f}$ does not have constant tangent kernel, while the tangent kernel of $f$ is constant. This implies that the constancy of the tangent kernel cannot be explained in terms of the magnitude of the parameter change from initialization, e.g., "lazy training". Instead, it depends on the structural properties of the network, such as the linearity of the output layer. Indeed, as we discuss next, when the output layer is non-linear, the Hessian norm of $\tilde{f}$ no longer decreases with the network width.

**Hessian matrix of $\tilde{f}$.** The Hessian matrix of $\tilde{f}$ is

$$\tilde{H} := \frac{\partial^2 \tilde{f}}{\partial \mathbf{w}^2} = \phi''(f)\nabla_{\mathbf{w}}f(\nabla_{\mathbf{w}}f)^T + \phi'(f)H, \tag{13}$$

where $H$ is the Hessian matrix of model $f$. Hence, the spectral norm satisfies

$$\|\tilde{H}\| \geq |\phi''(f)| \cdot \|\nabla_{\mathbf{w}}f\|^2 - |\phi'(f)| \cdot \|H\|. \tag{14}$$

Since, as we already know, $\lim_{m\to\infty} \|H\| = 0$, the second term vanishes in the infinite width limit. However, the first term is always of order $\Theta(1)$, as long as $\phi$ is not linear. Hence, $\|\tilde{H}\| = \Omega(1)$, compared to $\|H\| = \tilde{O}(1/\sqrt{m})$ for networks in Section 3 and does not vanish as $m \to \infty$.

**Wide neural networks with bottleneck.** Here, we show another example of neural networks that does not have constant tangent kernel by breaking the $\tilde{O}(1/\sqrt{m})$-scaling of Hessian spectral norm. Consider a neural network with fully connected layers. Here, we assume all the hidden layers are arbitrarily wide, except one layer, $l_b \neq L$, has a narrow width. For simplicity, let the bottleneck width $m_b = 1$. Now, the $(l_b + 1)$-th fully connected layer, Eq.(9), reduces to

$$\alpha^{(l_b+1)} = \sigma(\mathbf{w}^{(l_b+1)}\alpha^{(l_b)}),$$

with $\alpha^{(l_b)} \in \mathbb{R}$ and $\mathbf{w}^{(l_b+1)} \in \mathbb{R}^m$. In this case, the vector $\nabla_{\alpha^{(l_b)}}f$ becomes scalar and the $\infty$-norm $\|\nabla_{\alpha^{(l_b)}}f\|_\infty$ reduces to $|\nabla_{\alpha^{(l_b)}}f|$ which is of the order $\tilde{O}(1)$, not satisfying Condition (b) of

Theorem 3.1. Specifically, at the bottleneck, $\nabla_{\alpha^{(l_b)}} f = \sigma'(\mathbf{w}^{(l_b+1)} \alpha^{(l_b)}) \mathbf{w}^{(l_b+1)} \nabla_{\alpha^{(l_b+1)}} f$. Hence, at initialization $\mathbf{W}_0$,

$$\left| \nabla_{\alpha^{(l_b)}} f \right| = \left| \sigma'(\mathbf{w}_0^{(l_b+1)} \alpha^{(l_b)}) \right| \cdot \left| \mathbf{w}_0^{(l_b+1)} \nabla_{\alpha^{(l_b+1)}} f \right| = O\left( \left| \mathbf{w}_0^{(l_b+1)} \nabla_{\alpha^{(l_b+1)}} f \right| \right).$$

Using the same argument as in Section 3, we have $\| \nabla_{\alpha^{(l_b+1)}} f \|_\infty = \tilde{O}(1/\sqrt{m})$, i.e., $(\nabla_{\alpha^{(l_b+1)}} f)_i = \tilde{O}(1/\sqrt{m})$ for each $i \in [m]$, for the layer above the bottleneck. Since $\mathbf{w}_0^{(l_b+1)} \sim \mathcal{N}(0, I)$, the inner product $\mathbf{w}_0^{(l_b+1)} \nabla_{\alpha^{(l_b+1)}} f$, as a scalar, is of the order $\tilde{O}(1)$. Hence, $\| \nabla_{\alpha^{(l_b)}} f \|_\infty = \tilde{O}(1)$, not satisfying Condition (b) of Theorem 3.1, suggesting a non-constant tangent kernel during training. In Section 6, we empirically verify this finding.

In table 1, we summarize the key findings of this section and compare them with the case of neural networks with linear output layer.

| Network | Hessian norm | NTK | Trans. to linearity (constant NTK)? |
|---|---|---|---|
| linear output layer | $\tilde{O}(1/\sqrt{m})$ | $\Theta(1)$ | **Yes** |
| nonlinear output layer | $\tilde{O}(1)$ | $\Theta(1)$ | **No** |
| bottleneck | $\tilde{O}(1)$ | $\Theta(1)$ | **No** |

Table 1: Scaling of Hessian spectral norms of the models: linear output layer, non-linear output layer and bottleneck. Note: transition to linearity = constant tangent kernel, in the infinite width limit.

# 5 Optimization of wide neural networks

A number of recent analyses show convergence of gradient descent for wide neural networks [6, 7, 1, 19, 3, 11, 4]. While an extended discussion of optimization is beyond the scope of this work, we refer the interested reader to our separate paper [14]. The goal of this section is to clarify the important difference between the (near-)linearity of large models and convergence of optimization by gradient descent. It is easy to see that a wide model undergoing the "transition to linearity" can be optimized by gradient descent if its tangent kernel is well-conditioned at the initialization point. The dynamics of such a model will be essentially the same as for a linear model, an observation originally made in [10].

However near-linearity or, equivalently, near-constancy of the tangent kernel is not necessary for successful optimization. What is needed is that the tangent kernel is well-conditioned along the optimization path, a far weaker condition.

For a specific example, consider the non-linear output layer neural network $\tilde{f} = \phi(f)$, as defined in Eq. (11). As is shown in Section 4, this network does not have constant tangent kernel, even when the network width is arbitrarily large. The following theorem states that fast convergence of gradient descent still holds (also see Section 6 for empirical verification).

**Theorem 5.1.** *Suppose the non-linear function $\phi(\cdot)$ satisfies $|\phi'(z)| \geq \rho > 0, \forall z \in \mathbb{R}$, and the network width $m$ is sufficiently large. Then, with high probability of the random initialization, there exists constant $\mu > 0$, such that the gradient descent, with a small enough step size $\eta$, converges to a global minimizer of the square loss function $\mathcal{L}(\mathbf{w}) = \frac{1}{2} \sum_{i=1}^n (\tilde{f}(\mathbf{w}; \mathbf{x}_i) - y_i)^2$ with an exponential convergence rate:*

$$\mathcal{L}(\mathbf{w}_t) \leq (1 - \eta \mu \rho^2)^t \mathcal{L}(\mathbf{w}_0). \tag{15}$$

The analysis is based on the following reasoning. Convergence of gradient descent methods relies on the condition number of the tangent kernel (see [14]). It is not difficult to see that if the original model $f$ has a well conditioned tangent kernel, then the same holds for $\tilde{f} = \phi(f)$ as long as the the derivative of the activation function $\phi'$ is separated from zero. Since the tangent kernel of $f$ is not degenerate, the conclusion follows. The technical result is a consequence of Corollary 8.1 in [14].

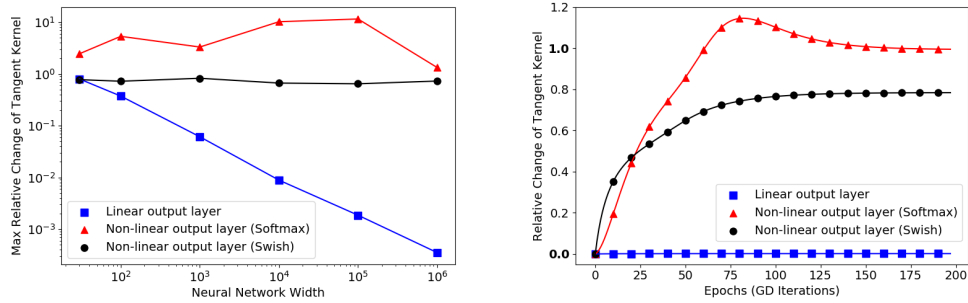

Figure 1: Neural networks with *non-linear* output layer vs. with *linear* output layer. Left panel (log scale): max change of tangent kernel $\Delta K$ from initialization to convergence w.r.t. the width $m$. Right panel: Evolution of tangent kernel change $\Delta K(t)$ as a function of epoch, width $m = 10^4$.

## 6 Numerical Verification

We conduct experiments to verify the non-constancy of tangent kernels for certain types of wide neural networks, as theoretically observed in Section 4.

Specifically, we use gradient descent to train each neural network described below on a synthetic data until convergence. We compute the following quantity to measure the max (relative) change of tangent kernel from initialization to convergence: $\Delta K := \sup_{t>0} \|K(\mathbf{w}_t) - K(\mathbf{w}_0)\|_F / \|K(\mathbf{w}_0)\|_F$. For a network that has a nearly constant tangent kernel during training, $\Delta K$ is expected to be close to 0, while a network with a non-constant tangent kernel, $\Delta K$ should be $\Omega(1)$. Detailed experimental setup and data description are given in Appendix C.

**Wide neural networks with non-linear output layers.** We consider a shallow (i.e., with one hidden layer) neural network $\tilde{f}$ of the type in Eq.(11) that has a softmax layer or swish [15] activation on the output. As a comparison, we consider a neural network $f$ that has the same structure as $\tilde{f}$, except that the output layer is linear.

We report the change of tangent kernels $\Delta K$ of $\tilde{f}$ and $f$, at different network width $m = \{30, 10^2, 10^3, 10^4, 10^5, 10^6\}$. The results are plotted in the left panel of Figure 1. We observe that, as the network width increases, the tangent kernel of $f$, which has a linear output layer, tends to be constant during training. However, the tangent kernel of $\tilde{f}$ which has a non-linear (softmax or swish) output layer, always takes significant change, even if the network width is large.

In Figure 1, right panel, we demonstrate the evolution of tangent kernel with respect to the training time for a very wide neural network (width $m = 10^4$). We see that, for the neural network with a non-linear output layer, tangent kernel changes significantly from initialization, while tangent kernel of the linear output network is nearly unchanged during training.

**Wide neural networks with a bottleneck.** We consider a fully connected neural network with 3 hidden layers and a linear output layer. The second hidden layer, i.e., the bottleneck layer, has a width $m_b$ which is typically small, while the width $m$ of the other hidden layers are typically very large, $m = 10^4$ in our experiment. For different bottleneck width $m_b = \{3, 5, 10, 50, 100, 500, 1000\}$, we train the network on a synthetic data using gradient descent until convergence, and compute $\Delta K$.

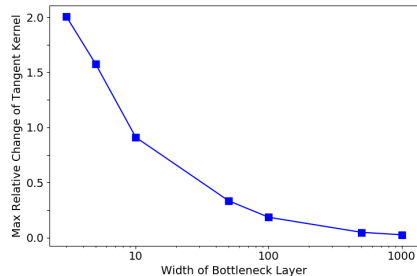

The change of tangent kernels for different bottleneck width is shown in Figure 2. We can see that a narrow bottleneck layer in a wide neural network prevent the neural tangent kernel from being constant during training. As expected, increasing the width of the bottleneck layer, makes the change of the tangent kernel smaller.

Figure 2: Networks with bottleneck. Relative change of tangent kernel from initialization to convergence, as a function of the bottleneck width.

## Broader impact

Our work concentrates on the theoretical aspects of neural networks. We believe that better understanding of the mathematical structures underlying their empirical success is a necessary component of further progress on their practical applications.

## Acknowledgements

The authors acknowledge support from NSF (IIS-1815697) and a Google Faculty Research Award. The GPU used for the experiments was donated by Nvidia.

## Footnotes

[1]After this NeurIPS submission, we developed a more general and more principled mathematical analysis of the transition to linearity phenomenon. Please see the details in: https://arxiv.org/abs/2010.01092

[2]While it is a known mathematical fact, [8, 16], we were not able to find it in the neural network literature, as the discussion is usually concerned with the dynamics of optimization controlled by the tangent kernel.

[3]We note that all standard architectures satisfy (b). In fact (b) can be relaxed to allow dependence on a small number of linear preactivations with essentially same analysis.

[4]If $\phi$ is linear, the term $A$ is identically zero.

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
