[Supplementary Material]

# Supplementary Material: Appendices

## A Can the transition to linearity be explained by model rescaling?

The work [5] introduced the term "lazy training" and proposed a mechanism for the constancy of the tangent kernel based on rescaling the model. While, as shown in [5], model rescaling can lead to lazy training, as we discuss below, it does not explain the phenomenon of constant tangent kernel in the setting of the original paper [10] and consequent works.

Specifically, [5] provides the following criterion for the near constancy of the tangent kernel (using their notation):

$$\kappa_f(\mathbf{w}_0) := \underbrace{\|f(\mathbf{w}_0) - y\|}_{\mathcal{A}} \underbrace{\frac{\|D^2 f(\mathbf{w}_0)\|}{\|Df(\mathbf{w}_0)\|^2}}_{\mathcal{B}} \ll 1, \tag{16}$$

Here $y$ is the ground truth label, $D^2 f(\mathbf{w}_0)$ is the Hessian of the model $f$ at initialization and $\|Df(\mathbf{w}_0)\|^2$ is the norm of the gradient, i.e., a diagonal entry of the tangent kernel.

The paper [5] shows that the model $f$ can be rescaled to satisfy the condition in Eq.(16) as follows. Consider a rescaled model $\alpha f$ with a scaling factor $\alpha \in \mathbb{R}, \alpha > 0$. Then the quantity $\kappa_{\alpha f}$ becomes

$$\kappa_{\alpha f}(\mathbf{w}_0) = \frac{1}{\alpha} \|\alpha f(\mathbf{w}_0) - y\| \frac{\|D^2 f(\mathbf{w}_0)\|}{\|Df(\mathbf{w}_0)\|^2} = \left\| f(\mathbf{w}_0) - \frac{y}{\alpha} \right\| \frac{\|D^2 f(\mathbf{w}_0)\|}{\|Df(\mathbf{w}_0)\|^2}. \tag{17}$$

Assuming that $f(\mathbf{w}_0) = 0$ and choosing a large $\alpha$, forces $\kappa_{\alpha f} \ll 1$, by rescaling the factor $\mathcal{A}$ to be small, while keeping $\mathcal{B}$ unchanged.

While rescaling the model, together with the important assumption of $f(\mathbf{w}_0) = 0$, leads to a lazy training regime, we point out that it is not the same regime as observed in the original work [10] and followup papers such as [13, 6] and also different from practical neural network training, since we usually have $\mathcal{A} = \|f(\mathbf{w}_0) - y\| = \Theta(1)$ in these settings. Specifically:

- The assumption of $f(\mathbf{w}_0) = 0$ is necessary for the rescaled models in [5] to have $\mathcal{A} \ll 1$. Yet, the networks, such as those analyzed in [10], are initialized so that $f(\mathbf{w}_0) = \Theta(1)$.

- From Eq.(17), we see that rescaling the model $f$ by $\alpha$ is equivalent to rescaling the ground truth label $y$ by $1/\alpha$ without changing the model (this can also be seen from the loss function, cf. Eq.(2) of [5]). When $\alpha$ is large, the rescaled label $y/\alpha$ is close to zero. However, no such rescaling happens in practice or in works, such as [10, 13, 6]. The training dynamics of the model with the label $y/\alpha$ does not generally match the dynamics of the original problem with the label $y$ and will result in a different solution.

Since $\mathcal{A} = \Theta(1)$, in the NTK setting and many practical settings, to satisfy the criterion in Eq.(16), the model needs to have $\mathcal{B} = \|D^2 f(\mathbf{w}_0)\| / \|Df(\mathbf{w}_0)\|^2 \ll 1$. In fact, we note that the analysis of 2-layer networks in [5] uses a different argument, not based on model rescaling. Indeed, as we show in this work, $\mathcal{B}$ is small for a broad class of wide neural networks with linear output layer, due to a vanishing norm of the Hessian as the width of the network increases.

In summary, the rescaled models satisfy the criterion, $\kappa \ll 1$, by scaling the factor $\mathcal{A}$ to be small, while the neural networks, such as the ones considered in the original work [10], satisfy this criterion by having $\mathcal{B} \ll 1$, while $\mathcal{A} = \Theta(1)$.

## B Other Parameterization Strategies

Throughout the paper, our analysis is based on the NTK prameterization [10], under which the constancy of tangent kernel is originally observed. In this section, we show that different parameterization strategies (e.g., LeCun initialization [LeCun et al.][5]: $w_{0;i}^{(l)} \sim \mathcal{N}(0, 1/m)$) do not change our conclusions. Specifically, we show that, compared to the NTK prameterization, a different

parameterization strategy only rescales the tangent kernel $K$ and the spectral norm of the Hessian $\|H\|$ by the same factor, hence the ratio between tangent kernel $K$ and Hessian spectral norm keeps the same and $\|H\| = o(\|K\|)$ still holds.

Recall that we initialize the parameters $\mathbf{W} = \{\mathbf{w}^{(1)}, \mathbf{w}^{(2)}, \cdots, \mathbf{w}^{(L)}, \mathbf{w}^{(L+1)} := \mathbf{v}\}$ of the general form of a deep neural network $f$, Eq.(7) by a standard Gaussian, i.e. $w_i^{(l)} \sim \mathcal{N}(0,1)$. If we apply another parameterization strategy $\overline{\mathbf{W}}$ here, for example, $\overline{w}_i^{(l)} \sim \mathcal{N}(0, \sigma_m^2)$, where $\sigma_m$ can be a function of $m$, we can see every $\overline{w}_i^{(l)} = \sigma_m w_i^{(l)}$ where $w_i^{(l)} \sim \mathcal{N}(0,1)$.

Hence each layer function becomes

$$\alpha^{(l)} = \phi_l \left( \frac{1}{\sigma_m} \overline{\mathbf{w}}^{(l)}; \alpha^{(l-1)} \right). \tag{18}$$

In this case, the gradient of the model $f$ w.r.t. the weights of layer $l$ is

$$\frac{\partial f}{\partial \overline{\mathbf{w}}^{(l)}} = \frac{\partial \mathbf{w}^{(l)}}{\partial \overline{\mathbf{w}}^{(l)}} \frac{\partial f}{\partial \mathbf{w}^{(l)}} = \frac{1}{\sigma_m} \frac{\partial f}{\partial \mathbf{w}^{(l)}}. \tag{19}$$

And by the same reason, the Hessian of the model f w.r.t. the weights of layer $l_1$ and $l_2$ is

$$\frac{\partial^2 f}{\partial \overline{\mathbf{w}}^{(l_1)} \partial \overline{\mathbf{w}}^{(l_2)}} = \frac{1}{\sigma_m^2} \frac{\partial^2 f}{\partial \mathbf{w}^{(l_1)} \partial \mathbf{w}^{(l_2)}}. \tag{20}$$

Therefore, it's easy to see the ratio of the norm of the tangent kernel to the norm of the Hessian keeps the same:

$$\frac{\|K(\overline{\mathbf{W}})\|}{\|H(\overline{\mathbf{W}})\|} = \frac{\|\frac{1}{\sigma_m^2} K(\mathbf{W})\|}{\|\frac{1}{\sigma_m^2} H(\mathbf{W})\|} = \frac{\|K(\mathbf{W})\|}{\|H(\mathbf{W})\|}. \tag{21}$$

**Example: LeCun initialization/parameterization.** In many practical machine learning tasks, it is popular to use the LeCun initialization/parameterization: each individual parameter $(W_0^{(l)})_{ij} \sim \mathcal{N}(0, \frac{1}{m})$, while there is no factor $1/\sqrt{m}$ in the definition of the layer function, e.g., for fully connected layers

$$\alpha^{(l+1)} = \sigma(W^{(l)} \alpha^{(l)}). \tag{22}$$

In this setting, the factor $\sigma_m = 1/\sqrt{m}$. Then, by the analysis above, we see that

$$\|K\| = O(m), \quad \|H\| = O(\sqrt{m}) = o(\|K\|). \tag{23}$$

It is also interesting to note that, for the parameter change $\mathbf{w}^* - \mathbf{w}_0$, the Euclidean norm

$$\|\mathbf{w}^* - \mathbf{w}_0\| = \Theta(1/\sqrt{m}). \tag{24}$$

## C   Experimental Setup

**Dataset.** We use a synthetic dataset of size $N = 60$ which contains $C = 3$ classes. Each data point $(x, y)$ is sampled as follows: label $y$ is randomly sampled from $\{0, 1, 2\}$ with equal probability; given $y$, $x$ is drawn from the following distribution:

$$x \sim \begin{cases} \mathcal{N}(0, 1), & \text{if } y = 0; \\ \mathcal{N}(10, 1), & \text{if } y = 1; \\ \mathcal{N}(-10, 1), & \text{if } y = 2. \end{cases} \tag{25}$$

We encode each $y_i \in \{0, 1, 2\}$ by a one-hot vector $\mathbf{y}_i \in \{0, 1\}^3$. And $\mathbf{y}_{i,j}$ means the $j$-th component of $\mathbf{y}_i$. We use this dataset for all the optimization tasks mentioned below.

### C.1   Wide neural networks with non-linear output layers

**Neural Networks.** In the experiments, we train three different neural networks:

- Neural network with a linear output layer

$$f(\mathbf{w}, V, \mathbf{b}; x) = \frac{1}{\sqrt{m}} V \sigma(\mathbf{w}x + \mathbf{b}), \tag{26}$$

where $\mathbf{w} \in \mathbb{R}^m$ and $\mathbf{b} \in \mathbb{R}^m$ are weights and biases for the first layer and $V \in \mathbb{R}^{3 \times m}$ are the weights for the output layer, and $\sigma(\cdot)$ is the ReLU activation function.

- Neural network with a softmax-activated (non-linear) output layer

$$\tilde{f}_1(\mathbf{w}, V, \mathbf{b}; x) = \mathrm{Softmax}(f(\mathbf{w}, V, \mathbf{b}; x)); \tag{27}$$

- Neural network with a swish-activated (non-linear) output layer

$$\tilde{f}_2(\mathbf{w}, V, \mathbf{b}; x) = \mathrm{Swish}(f(\mathbf{w}, V, \mathbf{b}; x)). \tag{28}$$

Here the swish activation function is defined as $\mathrm{Swish}(\mathbf{z}) = \mathbf{z} \odot (1 + \exp(-0.1 \cdot \mathbf{z}))^{-1}$, where $\odot$ is the element-wise multiplication.

**Optimization Tasks.** We combine the training of networks $f$ and $\tilde{f}_1$ together, by optimizing the following loss function:

$$\mathcal{L}_1(\mathbf{w}, \mathbf{v}, \mathbf{b}) = -\frac{1}{N} \sum_{i=1}^{N} \sum_{j=1}^{C} \mathbf{y}_{i,j} \cdot log((\tilde{f}_1(\mathbf{x}_i)_j). \tag{29}$$

In this combined training, networks $f$ and $\tilde{f}_1$ always have the same parameters during training, and the difference between $f$ and $\tilde{f}_1$ is the non-linearity on the output.

For the swish-activated network $\tilde{f}_2$, we minimize the square loss function:

$$\mathcal{L}_2(\mathbf{w}, \mathbf{v}, \mathbf{b}) = \frac{1}{N} \sum_{i=1}^{N} \|\tilde{f}_2(\mathbf{x}_i) - \mathbf{y}_i\|^2. \tag{30}$$

We use gradient descent to minimize the loss functions until convergence is achieved (i.e. loss less than $10^{-4}$). To measure the change of tangent kernels, we compute the max (relative) change of tangent kernel from initialization to convergence: $\Delta K := \sup_{t>0} \|K(\mathbf{w}_t) - K(\mathbf{w}_0)\|_F / \|K(\mathbf{w}_0)\|_F$. For each training, we take 10 independent runs and report the average $\Delta K$.

We compare the tangent kernel changes $\Delta K$ of $f$, $\tilde{f}_1$ and $\tilde{f}_2$, at a variety of network widths, $m = 30$, $10^2$, $10^3$, $10^4$, $10^5$, $10^6$.

### C.2  Wide neural networks with a bottleneck

**The Neural Network.**  In the experiment, we use a fully connected neural network with 3 hidden layers and a linear output layer. Its second hidden layer, i.e., the bottleneck layer has a width $m_b$, while the other hidden layers has a width $m$. Specifically, it is defined as:

$$f(\mathbf{W}; x) = \frac{1}{\sqrt{m}} W_4 \sigma \left( W_3 \frac{1}{\sqrt{m}} W_2 \sigma(W_1 x) \right), \tag{31}$$

where $W_1 \in \mathbb{R}^{m \times 1}$, $W_2 \in \mathbb{R}^{m_b \times m}$, $W_3 \in \mathbb{R}^{m \times m_b}$, $W_4 \in \mathbb{R}^{C \times m}$. Here we use ReLU as activation functions.

**Optimization Tasks.**  We minimize the cross entropy loss:

$$\mathcal{L}(\mathbf{W}) = -\frac{1}{N} \sum_{i=1}^{N} \sum_{j=1}^{C} \mathbf{y}_{i,j} \cdot log((\tilde{f}(\mathbf{x}_i)_j), \tag{32}$$

where we denote $\mathrm{Softmax}(f)$ by $\tilde{f}$. Here, we let the network width $m = 10^4$, and investigate on different bottleneck width $m_b \in \{3, 5, 10, 50, 100, 500, 1000\}$.

For each bottleneck width, we use gradient descent to minimize the loss functions until convergence is achieved (i.e. loss less than $10^{-4}$) and compute the max (relative) change of tangent kernel from initialization to convergence: $\Delta K := \sup_{t>0} \|K(\mathbf{w}_t) - K(\mathbf{w}_0)\|_F / \|K(\mathbf{w}_0)\|_F$. For each training, take 10 independent runs and report the average $\Delta K$.

# D   Proof for Proposition 2.1

*Proof.* Recall that the tangent kernel is defined as

$$K_{ij}(\mathbf{w}) = \nabla f(\mathbf{w}; \mathbf{x}_i)^T \nabla f(\mathbf{w}; \mathbf{x}_j), \quad \text{for any inputs } \mathbf{x}_i, \mathbf{x}_j \in \mathbb{R}^d. \tag{33}$$

**Linearity of $f$ in $\mathbf{w}$ $\Rightarrow$ constancy of tangent kernel.** Since $f$ is linear in $\mathbf{w}$, $\nabla_{\mathbf{w}} f(\mathbf{w}; \mathbf{x})$ is a constant vector in $\mathbb{R}^p$, for any given input $\mathbf{x}$. By the definition of the tangent kernel, each element $K_{ij}(\mathbf{w})$ is constant, for any inputs $\mathbf{x}_i, \mathbf{x}_j$.

**Constancy of tangent kernel $\Rightarrow$ linearity of $f$ in $\mathbf{w}$.** It suffices to prove for every input $\mathbf{x}_i$, function $f(\mathbf{w}; \mathbf{x}_i) : \mathbb{R}^p \to \mathbb{R}$ is linear in $\mathbf{w}$.

For a constant tangent kernel, each element $K_{ii}(\mathbf{w})$ is constant. Noting that $K_{ii}(\mathbf{w}) = \|\nabla_{\mathbf{w}} f(\mathbf{w}, \mathbf{x}_i)\|^2$, we have $\|\nabla_{\mathbf{w}} f(\mathbf{w}, \mathbf{x})\|$ is constant in $\mathbf{w}$, for all input $\mathbf{x}$.

The following arguments basically follow the idea from [8] (a more general result was shown in [16]).

To simplify the notation, in the rest of the proof, we hide the argument $\mathbf{x}$, and we use $f(\mathbf{w})$ to denote $f(\mathbf{w}; \mathbf{x})$.

Let $\|\nabla f(\mathbf{w})\| = c$. Consider the ordinary differential equation (ODE)

$$\frac{d\mathbf{w}(t)}{dt} = \nabla f(\mathbf{w}(t)),$$

where $\mathbf{w}(0) = \mathbf{w}_0 \in \mathbb{R}^p$ is the initial setting of the parameters. We have

$$\frac{df}{dt} = \langle \nabla f, \frac{d\mathbf{w}}{dt} \rangle = c^2,$$

and consequently

$$f(\mathbf{w}(t)) = c^2 t + f(\mathbf{w}_0). \tag{34}$$

For any $t_1, t_2$, since $\|\nabla f(\mathbf{w})\| = c$, we have

$$c^2 |t_1 - t_2| = |f(\mathbf{w}(t_1)) - f(\mathbf{w}(t_2))| \le c|\mathbf{w}(t_1) - \mathbf{w}(t_2)|,$$

but $|\mathbf{w}(t_1) - \mathbf{w}(t_2)| = |\int_{t_2}^{t_1} \|d\mathbf{w}(t)/dt\| dt| = c|t_1 - t_2|$, which indicates

$$\mathbf{w}(t) = t \nabla f(\mathbf{w}_0) + \mathbf{w}_0. \tag{35}$$

And in the following we show for any $\mathbf{v} \in \mathbb{R}^p$, if $f(\mathbf{v}) = f(\mathbf{w}_0)$, we have

$$\langle \nabla f(\mathbf{w}_0), \mathbf{v} - \mathbf{w}_0 \rangle = 0. \tag{36}$$

Given $t \ne 0$, let $c : [0, 1] \to \mathbb{R}^p$ be a differentiable curve joining $t\nabla f(\mathbf{w}_0) + \mathbf{w}_0$ and $\mathbf{v}$. By Eq. (34) and Eq. (35), we have

$$c^2 |t| = |f(\mathbf{w}(t)) - f(\mathbf{w}_0)| = |f(\mathbf{w}_0 + t\nabla f(\mathbf{w}_0)) - f(\mathbf{w}_0)|$$

$$= |\int_0^1 \langle \nabla f(c(s)), c'(s) \rangle ds|$$

$$\le \int_0^1 \|c'(s)\| ds$$

$$= \|\mathbf{v} - t\nabla f(\mathbf{w}_0) - \mathbf{w}_0\|.$$

It follows that

$$c^4 t^2 \le \|\mathbf{v} - \mathbf{w}_0\|^2 + t^2 + 2t\langle \mathbf{v} - \mathbf{w}_0, \nabla f(\mathbf{w}_0) \rangle.$$

Dividing by $t$ and taking $t$ to $\pm \infty$ allows us to have $\langle \nabla f(\mathbf{w}_0), \mathbf{v} - \mathbf{w}_0 \rangle = 0$.
Then we construct the level set

$$M_a = M = \{\mathbf{w} \in \mathbb{R}^p : f(\mathbf{w}) = a\}, \tag{37}$$

where $a \in \mathbb{R}$. And its tangent space at $\mathbf{w}$ is

$$T_{\mathbf{w}}M = \{\mathbf{w} + \mathbf{v} \in \mathbb{R}^p : \langle \mathbf{v}, \nabla f(\mathbf{w}) \rangle = 0\}. \tag{38}$$

By Eq.(36) we have $\langle \mathbf{v} - \mathbf{w}, \nabla f(\mathbf{w}) \rangle$ for all $\mathbf{v} \in M$ that satisfies $f(\mathbf{v}) = f(\mathbf{w})$. From Eq.(38) we can see $\mathbf{v} \in T_{\mathbf{w}}M$. Therefore $M \subset T_{\mathbf{w}}M$. By the fact that $M$ is a closed hypersurface, $M = T_{\mathbf{w}}M$ for all $\mathbf{w} \in M$.

Hence there exists a $\mathbf{w}' \in \mathbb{R}^p$ such that $\|\mathbf{w}'\| = 1$ and the level set Eq.(37) is equivalently defined as $M_a = \{\frac{a}{c}\mathbf{w}' + \mathbf{v}' : \langle \mathbf{v}', \mathbf{w}' \rangle = 0, \mathbf{v}' \in \mathbb{R}^p\}$ for all $a$. And we can construct a function $g : \mathbb{R} \to \mathbb{R}$ such that

$$g(t) = f(\mathbf{v}' + t\mathbf{w}'),$$

where $g'(t) = c$ for all $t$ which shows $f$ is linear.

$\square$

# E    Proof of Proposition 2.2

*Proof.* The model $f$, as a function of the parameters $\mathbf{w}$, can be written as the form of Taylor expansion with Lagrange remainder term:

$$f(\mathbf{w}) = f(\mathbf{w}_0) + \nabla_{\mathbf{w}} f(\mathbf{w}_0)^T(\mathbf{w} - \mathbf{w}_0) + \frac{1}{2}(\mathbf{w} - \mathbf{w}_0)^T H(\boldsymbol{\xi})(\mathbf{w} - \mathbf{w}_0), \tag{39}$$

for some $\boldsymbol{\xi}$ on the line segment joining $\mathbf{w}$ and $\mathbf{w}_0$. Then Euclidean norm of the gradient change is bounded by

$$\|\nabla_{\mathbf{w}} f(\mathbf{w}) - \nabla_{\mathbf{w}} f(\mathbf{w}_0)\| = \|H(\boldsymbol{\xi})(\mathbf{w} - \mathbf{w}_0)\| \le \|H(\boldsymbol{\xi})\| \cdot \|(\mathbf{w} - \mathbf{w}_0)\| \le \|H(\boldsymbol{\xi})\|R. \tag{40}$$

Hence, according to the definition of the tangent kernel, for any inputs $\mathbf{x}, \mathbf{z} \in \mathbb{R}^d$,

$$
\begin{aligned}
&|K_{(\mathbf{x},\mathbf{z})}(\mathbf{w}) - K_{(\mathbf{x},\mathbf{z})}(\mathbf{w}_0)| \\
\le\ & \|\nabla_{\mathbf{w}} f(\mathbf{w}; \mathbf{x}) - \nabla_{\mathbf{w}} f(\mathbf{w}_0; \mathbf{x})\| \cdot \|\nabla_{\mathbf{w}} f(\mathbf{w}; \mathbf{z})\| + \|\nabla_{\mathbf{w}} f(\mathbf{w}; \mathbf{z}) - \nabla_{\mathbf{w}} f(\mathbf{w}_0; \mathbf{z})\| \cdot \|\nabla f(\mathbf{w}_0; \mathbf{x})\| \\
\le\ & \|H(\boldsymbol{\xi})\|R(\|\nabla_{\mathbf{w}} f(\mathbf{w}_0; \mathbf{x})\| + \|\nabla_{\mathbf{w}} f(\mathbf{w}; \mathbf{z})\|).
\end{aligned}
$$

Since $f$ is smooth, the gradients $\nabla_{\mathbf{w}} f(\mathbf{w}_0)$ and $\nabla_{\mathbf{w}} f(\mathbf{w})$ are bounded. Therefore, $|K_{(\mathbf{x},\mathbf{z})}(\mathbf{w}) - K_{(\mathbf{x},\mathbf{z})}(\mathbf{w}_0)| = O(\epsilon R)$.  $\square$

# F    Proof of Theorem 3.1

*Proof.* The Hessian matrix $H$ of the neural network can be written as the following structure:

$$H = \begin{pmatrix} H^{(1,1)} & H^{(1,2)} & \cdots & H^{(1,L+1)} \\ H^{(2,1)} & H^{(2,2)} & \cdots & H^{(2,L+1)} \\ \vdots & \vdots & \ddots & \vdots \\ H^{(L+1,1)} & H^{(L+1,2)} & \cdots & H^{(L+1,L+1)} \end{pmatrix}. \tag{41}$$

Here, each Hessian block $H^{(l_1,l_2)} := \frac{\partial^2 f}{\partial \mathbf{w}^{(l_1)} \partial \mathbf{w}^{(l_2)}}$ is the second derivative of $f$ w.r.t. its weights of $l_1$-th and $l_2$-th layers, where we treat the final layer parameters $\mathbf{v}$ as $\mathbf{w}^{(L+1)}$.

The following lemma allows us to bound the Hessian spectral norm by the norms of its blocks (see proof in Appendix I.1).

**Lemma F.1.** *Spectral norm of a matrix $H$ (41) is upper bounded by the sum of the spectral norm of its blocks, i.e. $\|H\| \le \sum_{l_1,l_2} \|H^{(l_1,l_2)}\|$, $l_1, l_2 \in [L+1]$.*

By this lemma, it suffices to prove that, for each Hessian block $H^{(l_1,l_2)}, l_1, l_2 \in [L+1]$, the spectral norm $\|H^{(l_1,l_2)}\| = \tilde{O}(1/\sqrt{m})$. Since the Hessian matrix is symmetry, without loss of generosity, we assume $1 \le l_1 \le l_2 \le L+1$.

**Case 1:** $1 \le l_1 \le l_2 \le L$. By the chain rule, the gradient of the model $f$ w.r.t. the weights of layer $l$, can be written as

$$\frac{\partial f}{\partial \mathbf{w}^{(l)}} = \frac{\partial \phi_l}{\partial \mathbf{w}^{(l)}} \left( \prod_{l'=l+1}^{L} \frac{\partial \phi_{l'}}{\partial \alpha^{(l'-1)}} \right) \frac{1}{\sqrt{m}} \mathbf{v}. \tag{42}$$

Then, the Hessian block has the following expression:

$$H^{(l_1,l_2)}$$
$$= \frac{\partial^2 \phi_{l_1}}{(\partial \mathbf{w}^{(l_1)})^2} \frac{\partial f}{\partial \alpha_{l_1}} \cdot \mathbb{I}_{l_1=l_2} + \left( \frac{\partial \phi_{l_1}}{\partial \mathbf{w}^{(l_1)}} \prod_{l'=l_1+1}^{l_2-1} \frac{\partial \phi_{l'}}{\partial \alpha^{(l'-1)}} \right) \frac{\partial^2 \phi_{l_2}}{\partial \alpha^{(l_2-1)} \partial \mathbf{w}^{(l_2)}} \left( \frac{\partial f}{\partial \alpha^{(l_2)}} \right)$$
$$+ \sum_{l=l_2+1}^{L} \left( \frac{\partial \phi_{l_1}}{\partial \mathbf{w}^{(l_1)}} \prod_{l'=l_1+1}^{l-1} \frac{\partial \phi_{l'}}{\partial \alpha^{(l'-1)}} \right) \frac{\partial^2 \phi_l}{(\partial \alpha^{(l-1)})^2} \left( \frac{\partial \phi_{l_2}}{\partial \mathbf{w}^{(l_2)}} \prod_{l'=l_2+1}^{l} \frac{\partial \phi_{l'}}{\partial \alpha^{(l'-1)}} \right) \left( \frac{\partial f}{\partial \alpha^{(l)}} \right)$$

Hence, the spectral norm of Hessian block $H^{(l_1,l_2)}$ is bounded by

$$\left\| H^{(l_1,l_2)} \right\|$$
$$\le \left\| \frac{\partial^2 \phi_{l_1}}{(\partial \mathbf{w}^{(l_1)})^2} \right\|_{2,2,1} \left\| \frac{\partial f}{\partial \alpha^{(l_1)}} \right\|_{\infty} + \left\| \frac{\partial \phi_{l_1}}{\partial \mathbf{w}^{(l_1)}} \right\| \prod_{l'=l_1+1}^{l_2-1} \left\| \frac{\partial \phi_{l'}}{\partial \alpha^{(l'-1)}} \right\| \left\| \frac{\partial^2 \phi_{l_2}}{\partial \alpha^{(l_2-1)} \partial \mathbf{w}^{(l_2)}} \right\|_{2,2,1} \left\| \frac{\partial f}{\partial \alpha^{(l_2)}} \right\|_{\infty}$$
$$+ \sum_{l=l_2+1}^{L} \left\| \frac{\partial \phi_{l_1}}{\partial \mathbf{w}^{(l_1)}} \right\| \prod_{l'=l_1+1}^{l} \left\| \frac{\partial \phi_{l'}}{\partial \alpha^{(l'-1)}} \right\| \left\| \frac{\partial^2 \phi_l}{(\partial \alpha^{(l-1)})^2} \right\|_{2,2,1} \left\| \frac{\partial \phi_{l_2}}{\partial \mathbf{w}^{(l_2)}} \right\| \prod_{l'=l_2+1}^{l} \left\| \frac{\partial \phi_{l'}}{\partial \alpha^{(l'-1)}} \right\| \left\| \frac{\partial f}{\partial \alpha^{(l)}} \right\|_{\infty}.$$

Since the vector-valued layer function $\phi_l$ is Lipschitz continuous w.r.t. its input $\alpha$ and parameters $\mathbf{w}$, the spectral norms of "gradients" $\left\| \frac{\partial \phi_l}{\partial \alpha^{(l-1)}} \right\|$ and $\left\| \frac{\partial \phi_l}{\partial \mathbf{w}^{(l)}} \right\|$ are bounded, for all $l \in [L]$. By the condition (a) of the theorem, the norms of order 3 tensors, $\left\| \frac{\partial^2 \phi_l}{(\partial \mathbf{w}^{(l)})^2} \right\|_{2,2,1}$, $\left\| \frac{\partial^2 \phi_l}{(\partial \alpha^{(l-1)})^2} \right\|_{2,2,1}$ and $\left\| \frac{\partial^2 \phi_{l_2}}{\partial \alpha^{(l_2-1)} \partial \mathbf{w}^{(l_2)}} \right\|_{2,2,1}$ are of the order $\tilde{O}(1)$. In addition, by condition (b) of the theorem, the $\infty$-norms $\left\| \frac{\partial f}{\partial \alpha^{(l)}} \right\|_{\infty}, l \in [L]$, are of the order $\tilde{O}(1/\sqrt{m})$. Hence, we have

$$\left\| H^{(l_1,l_2)} \right\| = \tilde{O}\left( \frac{1}{\sqrt{m}} \right), \quad \forall l_1, l_2 \in [L]. \tag{43}$$

**Case 2:** $1 \le l_1 < l_2 = L + 1$. Using the gradient expression in Eq.(42), we have

$$H^{(l_1,L+1)} = \frac{1}{\sqrt{m}} \frac{\partial \phi_l}{\partial \mathbf{w}^{(l_1)}} \left( \prod_{l'=l+1}^{L} \frac{\partial \phi_{l'}}{\partial \alpha^{(l'-1)}} \right). \tag{44}$$

Hence,

$$\| H^{(l_1,L+1)} \| \le \frac{1}{\sqrt{m}} \left\| \frac{\partial \phi_l}{\partial \mathbf{w}^{(l_1)}} \right\| \prod_{l'=l+1}^{L} \left\| \frac{\partial \phi_{l'}}{\partial \alpha^{(l'-1)}} \right\| = \tilde{O}\left( \frac{1}{\sqrt{m}} \right). \tag{45}$$

**Case 3:** $l_1 = l_2 = L + 1$. In this case, the Hessian block $H^{(L+1,L+1)}$ is simply zero. Hence, the spectral norm is zero. $\qquad\square$

## G   Proofs of Theorem 3.2

The basic idea of the proof is to show that this fully connected neural network satisfies, within a ball of finite radius around the initialization $\mathbf{W}_0$, the conditions proposed in Theorem 3.1, as well as the Lipschitz continuity assumption. Then, by Theorem 3.1 we immediately get the desired results.

The fully connected neural network is defined in the following way:

$$\alpha^{(0)} = \mathbf{x},$$

$$\alpha^{(l)} = \sigma(\tilde{\alpha}^{(l)}), \quad \tilde{\alpha}^{(l)} = \frac{1}{\sqrt{m_{l-1}}} W^{(l)} \alpha^{(l-1)}, \quad \forall l \in [L]$$

$$f = \frac{1}{\sqrt{m_L}} \mathbf{v}^T \alpha^{(L)}, \tag{46}$$

where $m_0 = d$ which is the dimension of the input $\mathbf{x}$, and $m_l = m$ for all $l \in [L]$. The trainable parameters of this network are $\mathbf{W} := \{W^{(1)}, W^{(2)}, \cdots, W^{(L)}, W^{(L+1)} := \mathbf{v}\}$, and are initialized by the random Gaussian initialization, i.e., each parameter $(W_0^{(l)})_{ij} \sim \mathcal{N}(0,1), \forall l \in [L]$, and $v_{0,i} \sim \mathcal{N}(0,1), i,j \in [m]$. As the parameters $W^{(l)}$ of each layer are reshaped into matrices, the Euclidean norm of parameters becomes $\|\mathbf{W}\| := (\sum_{l=1}^{L+1} \|W^{(l)}\|_F^2)^{1/2}$, where $\|\cdot\|_F$ is the Frobenius norm of a matrix.

To make the presentation of the proof as simple as possible, we first make the following assumption about the initial parameters $\mathbf{W}_0$. Then we prove it in Lemma G.1 that the assumption is satisfied with high probability by the random Gaussian initialization.

**Assumption G.1.** We assume that there exists a constant $c_0 > 0$ such that, for all initial weight matrices/vector $W_0^{(l)}$, $\|W_0^{(l)}\| \leq c_0 \sqrt{m}$, where $l \in [L+1]$.

**Lemma G.1** (Spectral norms of initial weight matrices). *If the parameters are initialized as* $(W_0^{(l)})_{ij} \sim \mathcal{N}(0,1)$ *for all* $l \in [L+1]$ *and* $m > d$, *then, for each layer* $l \in [L+1]$, *we have with probability at least* $1 - 2\exp(-\frac{m}{2})$,

$$\|W_0^{(l)}\| \leq 3\sqrt{m}. \tag{47}$$

The proof is in Appendix I.2.

We further assume that, for the input $\mathbf{x} \in \mathbb{R}^d$, each component is bounded, i.e. $|x_i| \leq C_{\mathbf{x}}$, for some constant $C_{\mathbf{x}}$ and for all $i \in [d]$. This assumption covers most of the practical cases.

We prove the following lemma which states that the norm of the matrix $W^{(l)}$ keeps its order in a finite ball around the $W_0^{(l)}$.

**Lemma G.2.** *If* $\mathbf{W}_0$ *satisfies Assumption G.1, then for any* $\mathbf{W}$ *such that* $\|\mathbf{W} - \mathbf{W}_0\| \leq R$, *we have*

$$\|W^{(l)}\| \leq c_0 \sqrt{m} + R = O(\sqrt{m}), \quad \forall l \in [L+1]. \tag{48}$$

See the proof in Appendix I.3. The following lemma gives bounds on the Euclidean norm of the vector of hidden neurons for each layer.

**Lemma G.3.** *If* $\mathbf{W}_0$ *satisfies Assumption G.1, then, for any* $\mathbf{W}$ *such that* $\|\mathbf{W} - \mathbf{W}_0\| \leq R$, *we have, at all hidden layers*

$$\|\alpha^{(l)}(\mathbf{W})\| \leq L_\sigma^l (c_0 + R/\sqrt{m})^l \sqrt{m} C_{\mathbf{x}} + \sum_{i=1}^l L_\sigma^{i-1} (c_0 + R/\sqrt{m})^{i-1} \sigma(0) = O(\sqrt{m}), \quad \forall l \in [L].$$
$$\tag{49}$$

*Particularly, for the input layer,*

$$\|\alpha^{(0)}\| = \|\mathbf{x}\| \leq \sqrt{d} C_{\mathbf{x}} = O(1). \tag{50}$$

The proof is in Appendix I.4 .

Now, we are ready to prove the theorem. Specifically, we prove the following for the fully connected layer in the ball $B(\mathbf{W}_0, R)$: Lipschitz continuity, satisfying conditions (a) and (b) in Theorem 3.1 in Section G.1, G.2 and G.3, respectively. Then, the theorem is an immediate consequence of Theorem 3.1.

## G.1 Proofs of Lipschitz continuity

Note that the layer function is vector-valued. It suffices to prove that the first derivatives, which are matrices, have bounded spectral norms in the ball $B(\mathbf{W}_0, R)$.

**When $l = 2, 3, \cdots, L$.** Recall from Eq.(9) that, a fully connected layer $\phi_l$ is defined as, for $l = 2, 3, \cdots, L$:

$$\alpha^{(l)} = \phi_l(W^{(l)}; \alpha^{(l-1)}) = \sigma\left(\frac{1}{\sqrt{m}} W^{(l)} \alpha^{(l-1)}\right). \tag{51}$$

The term $\tilde{\alpha}^{(l)} := \frac{1}{\sqrt{m}} W^{(l)} \alpha^{(l-1)}$ is also known as preactivation.

Note that, in this case, the parameter vector $\mathbf{w}^{(l)}$ is reshaped to a $m \times m$ matrix $W^{(l)}$, except that $W^{(1)}$ is a $m \times d$ matrix. The first derivatives of $\phi_l$ are

$$(\nabla_\alpha \phi_l)_{i,j} = \frac{1}{\sqrt{m}} \sigma'(\tilde{\alpha}_i^{(l)}) W_{ij}^{(l)}, \tag{52}$$

$$(\nabla_\mathbf{w} \phi_l)_{i,jj'} = \frac{1}{\sqrt{m}} \sigma'(\tilde{\alpha}_i^{(l)}) \alpha_{j'}^{(l-1)} \mathbb{I}_{i=j}. \tag{53}$$

By the definition of spectral norm, $\|A\| = \sup_{\|\mathbf{v}\|=1} \|A\mathbf{v}\|$, we have, for all $2 \leq l \leq L$,

$$
\begin{aligned}
\|\nabla_\alpha \phi_l\|^2 &= \sup_{\|\mathbf{v}\|=1} \frac{1}{m} \sum_{i=1}^m \left(\sigma'(\tilde{\alpha}_i^{(l)}) W_{ij}^{(l)} v_j\right)^2 \\
&= \sup_{\|\mathbf{v}\|=1} \frac{1}{m} \|\Sigma'^{(l)} W^{(l)} \mathbf{v}\|^2 \\
&\leq \frac{1}{m} \|\Sigma'^{(l)}\|^2 \|W^{(l)}\|^2 \\
&\leq L_\sigma^2 (c_0 + R/\sqrt{m})^2 = O(1),
\end{aligned}
$$

where $\Sigma'^{(l)}$ is a diagonal matrix, with the diagonal entry $\Sigma_{ii}'^{(l)} = \sigma'(\tilde{\alpha}_i^{(l)})$. Similarly, we have

$$
\begin{aligned}
\|\nabla_\mathbf{w} \phi_l\|^2 &= \sup_{\|V\|_F=1} \frac{1}{m} \sum_{i=1}^m \left(\sum_{j,j'} \sigma'(\tilde{\alpha}_i^{(l)}) \alpha_{j'}^{(l-1)} \mathbb{I}_{i=j} V_{jj'}\right)^2 \\
&= \sup_{\|V\|_F=1} \frac{1}{m} \|\Sigma'^{(l)} V \alpha^{(l-1)}\|^2 \\
&\leq \frac{1}{m} \|\Sigma'^{(l)}\|^2 \|\alpha^{(l-1)}\|^2 \\
&\leq \left(L_\sigma^l (c_0 + R)^{l-1} C_\mathbf{x}\right)^2 = O(1). 
\end{aligned} \tag{54}
$$

We used Lemma G.2 and G.3 above.

**When $l = 1$.** The layer function is:

$$\alpha^{(1)} = \phi_1(W^{(1)}; \alpha^{(0)}) = \sigma\left(\frac{1}{\sqrt{d}} W^{(1)} \mathbf{x}\right). \tag{55}$$

In this layer, the input $\mathbf{x}$ is fixed (independent of trainable parameters) and not a dynamical variable. Hence, $\nabla_\alpha \phi_1$ is not an interesting object in our Hessian analysis[6].

For $\nabla_\mathbf{w} \phi_1$, we have (with a similar analysis as in Eq.(54)),

$$\|\nabla_\mathbf{w} \phi_1\|^2 \leq \frac{1}{d} \|\Sigma'^{(l)}\|^2 \|\alpha^{(0)}\|^2 \leq L_\sigma^2 C_\mathbf{x}^2 = O(1).$$

We see that both $\|\nabla_\alpha \phi_l\|$ and $\|\nabla_\mathbf{w} \phi_l\|$ are bounded, hence, the (vector valued) layer function of fully connected neural networks is Lipschitz continuous.

## G.2 Proof of condition (a) in Theorem 3.1

*Proof.* We consider the first layer i.e. $l = 1$ and the rest of the layers i.e. $l = 2, 3, \cdots, L$ separately.

**When $l = 2, 3, \cdots, L$.** The second derivatives of the vector-valued layer function $\phi_l$, which are order 3 tensors, have the following expressions:

$$\left(\frac{\partial^2 \phi_l}{(\partial \alpha^{(l-1)})^2}\right)_{i,j,k} = \frac{1}{m}\sigma''(\tilde{\alpha}_i^{(l)})W_{ij}^{(l)}W_{ik}^{(l)}, \tag{56}$$

$$\left(\frac{\partial^2 \phi_l}{\partial \alpha^{(l-1)}\partial W^{(l)}}\right)_{i,j,kk'} = \frac{1}{m}\sigma''(\tilde{\alpha}_i^{(l)})W_{ij}^{(l)}\alpha_{k'}^{(l-1)}\mathbb{I}_{i=k}, \tag{57}$$

$$\left(\frac{\partial^2 \phi_l}{(\partial W^{(l)})^2}\right)_{i,jj',kk'} = \frac{1}{m}\sigma''(\tilde{\alpha}_i^{(l)})\alpha_{j'}^{(l-1)}\alpha_{k'}^{(l-1)}\mathbb{I}_{i=k=j}. \tag{58}$$

By the definition of the $(2, 2, 1)$-norm for order 3 tensors, and Lemma G.2, we get

$$\begin{aligned}
\left\|\frac{\partial^2 \phi_l}{(\partial \alpha^{(l-1)})^2}\right\|_{2,2,1} &= \sup_{\|\mathbf{v}_1\|=\|\mathbf{v}_2\|=1} \frac{1}{m}\sum_{i=1}^{m}\left|\sigma''(\tilde{\alpha}_i^{(l)})(W^{(l)}\mathbf{v}_1)_i(W^{(l)}\mathbf{v}_2)_i\right| \\
&\leq \sup_{\|\mathbf{v}_1\|=\|\mathbf{v}_2\|=1} \frac{1}{m}\beta_\sigma\sum_{i=1}^{m}\left|(W^{(l)}\mathbf{v}_1)_i(W^{(l)}\mathbf{v}_2)_i\right| \\
&\leq \sup_{\|\mathbf{v}_1\|=\|\mathbf{v}_2\|=1} \frac{1}{2m}\beta_\sigma\sum_{i=1}^{m}(W^{(l)}\mathbf{v}_1)_i^2 + (W^{(l)}\mathbf{v}_2)_i^2 \\
&\leq \frac{1}{2m}\beta_\sigma\sup_{\|\mathbf{v}_1\|=\|\mathbf{v}_2\|=1}(\|W^{(l)}\mathbf{v}_1\|^2 + \|W^{(l)}\mathbf{v}_2\|^2) \\
&\leq \frac{1}{2m}\beta_\sigma(\|W^{(l)}\|^2 + \|W^{(l)}\|^2) \\
&\leq \beta_\sigma(c_0 + R/\sqrt{m})^2 = O(1). \tag{59}
\end{aligned}$$

Similarly, by using Lemma G.2 and Lemma G.3, we have,

$$\begin{aligned}
\left\|\frac{\partial^2 \phi_l}{\partial \alpha^{(l-1)}\partial W^{(l)}}\right\|_{2,2,1} &= \sup_{\|\mathbf{v}_1\|=\|V_2\|_F=1} \frac{1}{m}\sum_{i=1}^{m}\left|\sigma''(\tilde{\alpha}_i^{(l)})(W^{(l)}\mathbf{v}_1)_i(V_2\alpha^{(l)})_i\right| \\
&\leq \sup_{\|\mathbf{v}_1\|=\|V_2\|_F=1} \frac{1}{2m}\beta_\sigma(\|W^{(l)}\mathbf{v}_1\|^2 + \|V_2\alpha^{(l-1)}\|^2) \\
&\leq \frac{1}{2m}\beta_\sigma(\|W^{(l)}\|^2 + \|\alpha^{(l-1)}\|^2) \\
&\leq \frac{\beta_\sigma}{2}(c_0 + R/\sqrt{m})^2 + \frac{\beta_\sigma}{2}L_\sigma^{2l-2}(c_0 + R/\sqrt{m})^{(2l-2)}C_\mathbf{x}^2 = O(1).
\end{aligned}$$

And

$$\begin{aligned}
\left\|\frac{\partial^2 \phi_l}{(\partial W^{(l)})^2}\right\|_{2,2,1} &= \sup_{\|V_1\|_F=\|V_2\|_F=1} \frac{1}{m}\sum_{i=1}^{m}\left|\sigma''(\tilde{\alpha}_i^{(l)})(V_1\alpha^{(l-1)})_i(V_2\alpha^{(l-1)})_i\right| \\
&\leq \sup_{\|V_1\|_F=\|V_2\|_F=1} \frac{1}{2m}\beta_\sigma(\|V_1\alpha^{(l-1)}\|^2 + \|V_2\alpha^{(l-1)}\|^2) \\
&\leq \frac{1}{2m}\beta_\sigma(\|\alpha^{(l-1)}\|^2 + \|\alpha^{(l-1)}\|^2) \\
&\leq \beta_\sigma L_\sigma^{2l-2}(c_0 + R/\sqrt{m})^{2l-2}C_\mathbf{x}^2 = O(1). \tag{60}
\end{aligned}$$

**When $l = 1$.** As discussed in Section G.1, the input $\alpha^{(0)}$ is constant, we only need to analyze the tensor $\frac{\partial^2 \phi_l}{(\partial W^{(l)})^2}$ in this case. With a similar analysis as in Eq.(60), we have

$$\left\|\frac{\partial^2 \phi_1}{(\partial W^{(1)})^2}\right\|_{2,2,1} \leq \frac{1}{2d}\beta_\sigma(\|\alpha^{(0)}\|^2 + \|\alpha^{(0)}\|^2) \leq \beta_\sigma C_\mathbf{x}^2 = O(1). \tag{61}$$

$\square$

### G.3 Proof of condition (b) in Theorem 3.1

*Proof.* First of all, we present a few useful facts, Lemma G.4-G.6 that will be used during the proof. The proofs of the following lemmas are in Appendix I.5-I.7.

We first show that each activation of the hidden layers is bounded at initialization, with high probability.

**Lemma G.4.** *For any $l \in [L]$, given $i \in [m]$, with probability at least $1 - 2e^{-c_\alpha^{(l)} \ln^2(m)}$ for some constant $c_\alpha^{(l)} > 0$, $|\alpha_i^{(l)}| = \tilde{O}(1)$ at initialization.*

Define vector $\mathbf{b}^{(l)} := \nabla_{\alpha^{(l)}} f \in \mathbb{R}^m$ for $l \in [L]$. And we use $\mathbf{b}_0$ to denote $\mathbf{b}$ at initialization. Specifically, $\mathbf{b}^{(l)}$ takes the following form:

$$\mathbf{b}^{(l)} = \prod_{l'=l+1}^{L} \left( \frac{1}{\sqrt{m}} (W^{(l')})^T \Sigma'^{(l')} \right) \frac{1}{\sqrt{m}} \mathbf{v}, \tag{62}$$

where $\Sigma'^{(l')}$ is a diagonal matrix, with $(\Sigma'^{(l')})_{ii} = \sigma'(\tilde{\alpha}_i^{(l')})$.

The following lemma gives an upper bound to norms of $\mathbf{b}^{(l)}$ in the ball $B(\mathbf{W}_0, R)$.

**Lemma G.5.** *If the initial parameters $\mathbf{W}_0$ of the multi-layer neural network $f(\mathbf{W})$ satisfies Assumption G.1, then, for any $\mathbf{W}$ such that $\|\mathbf{W} - \mathbf{W}_0\| \leq R$, we have, at all hidden layers, i.e., $\forall l \in [L]$,*

$$\|\mathbf{b}^{(l)}\| \leq L_\sigma^{L-l} (c_0 + R/\sqrt{m})^{L-l+1}. \tag{63}$$

*In particular, at initialization,*

$$\|\mathbf{b}_0^{(l)}\| \leq L_\sigma^{L-l} c_0^{L-l+1}. \tag{64}$$

We proceed to show all the components of $\mathbf{b}_0^{(l)}$ are of order $\tilde{O}(\frac{1}{\sqrt{m}})$ with high probability.

**Lemma G.6.** *With probability at least $1 - me^{-c_b^{(l)} \ln^2(m)}$ for some constant $c_b^{(l)} > 0$, $\|\mathbf{b}_0^{(l)}\|_\infty = \tilde{O}(1/\sqrt{m})$.*

Now we show besides at initialization, $\|\mathbf{b}\|_\infty$ is of order $\tilde{O}(\frac{1}{\sqrt{m}})$ in the ball $B(\mathbf{W}_0, R)$ with high probability, i.e. the condition (b) in Theorem 3.1. Technically, we bound the difference of the $\infty$-norm by the difference of 2-norm.

First of all, we prove, by induction, the following claim: for all $l \in [L]$,

$$\|\mathbf{b}^{(l)} - \mathbf{b}_0^{(l)}\| = \tilde{O}\left(\frac{1}{\sqrt{m}}\right). \tag{65}$$

In the base case, we consider $l = L$. We have

$$\mathbf{b}^{(L)} = \frac{1}{\sqrt{m}} \mathbf{v},$$

$$\mathbf{b}_0^{(L)} = \frac{1}{\sqrt{m}} \mathbf{v}_0.$$

Hence,

$$\|\mathbf{b}^{(L)} - \mathbf{b}_0^{(L)}\| = \frac{1}{\sqrt{m}} \|\mathbf{v} - \mathbf{v}_0\| \leq \frac{1}{\sqrt{m}} \|\mathbf{W} - \mathbf{W}_0\| \leq \frac{1}{\sqrt{m}} R. \tag{66}$$

Now, suppose that $\|\mathbf{b}^{(l)} - \mathbf{b}_0^{(l)}\| = \tilde{O}(\frac{1}{\sqrt{m}})$. Then

$$
\begin{aligned}
\left\|\mathbf{b}^{(l-1)} - \mathbf{b}_0^{(l-1)}\right\| &= \frac{1}{\sqrt{m}}\left\|(W^{(l)})^T\Sigma'^{(l)}\mathbf{b}^{(l)} - (W_0^{(l)})^T\Sigma_0'^{(l)}\mathbf{b}_0^{(l)} + (W_0^{(l)})^T\Sigma'^{(l)}\mathbf{b}_0^{(l)}\right. \\
&\quad \left. + (W_0^{(l)})^T\Sigma'^{(l)}\mathbf{b}^{(l)} - (W_0^{(l)})^T\Sigma'^{(l)}\mathbf{b}_0^{(l)} - (W_0^{(l)})^T\Sigma'^{(l)}\mathbf{b}^{(l)}\right\| \\
&= \frac{1}{\sqrt{m}}\left\|\left((W^{(l)})^T - (W_0^{(l)})^T\right)\Sigma'^{(l)}\mathbf{b}^{(l)} + (W_0^{(l)})^T\left(\Sigma'^{(l)} - \Sigma_0'^{(l)}\right)\mathbf{b}_0^{(l)}\right. \\
&\quad \left. + (W_0^{(l)})^T\Sigma'^{(l)}\left(\mathbf{b}^{(l)} - \mathbf{b}_0^{(l)}\right)\right\| \\
&\leq \frac{1}{\sqrt{m}}\left\|W^{(l)} - W_0^{(l)}\right\|_2\|\Sigma'^{(l)}\|\|\mathbf{b}^{(l)}\| + \frac{1}{\sqrt{m}}\|W_0^{(l)}\|\left\|\left(\Sigma'^{(l)} - \Sigma_0'^{(l)}\right)\mathbf{b}_0^{(l)}\right\| \\
&\quad + \frac{1}{\sqrt{m}}\|W_0^{(l)}\|\|\Sigma'^{(l)}\|\left\|\mathbf{b}^{(l)} - \mathbf{b}_0^{(l)}\right\|,
\end{aligned} \tag{67}
$$

where $\Sigma'^{(l)}$ is a diagonal matrix, with $(\Sigma'^{(l)})_{ii} = \sigma'(\tilde{\alpha}_i^{(l)})$.

To bound the second additive term above, we need the following inequality:

$$
\begin{aligned}
&\|\tilde{\alpha}^{(l)}(\mathbf{W}) - \tilde{\alpha}^{(l)}(\mathbf{W}_0)\| \\
=\ & \left\|\frac{1}{\sqrt{m}}W^{(l)}\alpha^{(l-1)}(\mathbf{W}) - \frac{1}{\sqrt{m}}W_0^{(l)}\alpha^{(l-1)}(\mathbf{W}_0)\right\| \\
\leq\ & \frac{1}{\sqrt{m}}\|W_0^{(l)}\|\cdot L_\sigma \cdot \|\tilde{\alpha}^{(l-1)}(\mathbf{W}) - \tilde{\alpha}^{(l-1)}(\mathbf{W}_0)\| + \frac{1}{\sqrt{m}}\|W^{(l)} - W_0^{(l)}\|\|\alpha^{(l-1)}(\mathbf{W})\| \\
\leq\ & c_0 L_\sigma\|\tilde{\alpha}^{(l-1)}(\mathbf{W}) - \tilde{\alpha}^{(l-1)}(\mathbf{W}_0) + \frac{1}{\sqrt{m}}\|W^{(l)} - W_0^{(l)}\|\|\alpha^{(l-1)}(\mathbf{W})\| \\
=\ & c_0 L_\sigma\|\tilde{\alpha}^{(l-1)}(\mathbf{W}) - \tilde{\alpha}^{(l-1)}(\mathbf{W}_0)\| + O(1),
\end{aligned}
$$

where the last equality is the result of Lemma G.3 that $\|\alpha^{(l-1)}\| = O(\sqrt{m})$.

Recursively applying the above equation, since $\|\tilde{\alpha}^{(1)}(\mathbf{W}) - \tilde{\alpha}^{(1)}(\mathbf{W}_0)\| \leq \frac{R}{\sqrt{d}}C_{\mathbf{x}}$, we have

$$
\|\tilde{\alpha}^{(l)}(\mathbf{W}) - \tilde{\alpha}^{(l)}(\mathbf{W}_0)\| = c_0^{l-1}L_\sigma^{l-1}\|\tilde{\alpha}^{(1)}(\mathbf{W}) - \tilde{\alpha}^{(1)}(\mathbf{W}_0)\| + O(1) = O(1). \tag{68}
$$

Also, note that $\Sigma'$ is a diagonal matrix, then, we have

$$
\begin{aligned}
\left\|\left[\Sigma'^{(l)} - \Sigma_0'^{(l)}\right]\mathbf{b}_0^{(l)}\right\| &= \sqrt{\sum_{i=1}^m (\mathbf{b}_0^{(l)})_i^2\left(\sigma'(\tilde{\alpha}_i^{(l)}(\mathbf{W})) - \sigma'(\tilde{\alpha}_i^{(l)}(\mathbf{W}_0))\right)^2} \\
&\leq \|\mathbf{b}_0^{(l)}\|_\infty\sqrt{\sum_{i=1}^m\left[\sigma'(\tilde{\alpha}_i^{(l)}(\mathbf{W})) - \sigma'(\tilde{\alpha}_i^{(l)}(\mathbf{W}_0))\right]^2} \\
&\leq \|\mathbf{b}_0^{(l)}\|_\infty\cdot\beta_\sigma\|\tilde{\alpha}^{(l)}(\mathbf{W}) - \tilde{\alpha}^{(l)}(\mathbf{W}_0)\| = \tilde{O}\left(\frac{1}{\sqrt{m}}\right),
\end{aligned} \tag{69}
$$

where we used Lemma G.5 and Eq.(68) in the last equality.

Now, insert Eq.(69) into Eq.(67), and apply Lemma G.5 and the induction hypothesis, then we have

$$
\begin{aligned}
\left\|\mathbf{b}^{(l-1)} - \mathbf{b}_0^{(l-1)}\right\| &\leq \frac{1}{\sqrt{m}}RL_\sigma^{L-l+1}(c_0 + R/\sqrt{m})^{L-l+1} + \frac{1}{\sqrt{m}}c_0\sqrt{m}\left\|\left[\Sigma'^{(l)} - \Sigma_0'^{(l)}\right]\mathbf{b}_0^{(l)}\right\| \\
&\quad + c_0 L_\sigma\left\|\mathbf{b}^{(l)} - \mathbf{b}_0^{(l)}\right\| = \tilde{O}\left(\frac{1}{\sqrt{m}}\right).
\end{aligned} \tag{70}
$$

Thus, Eq.(65) holds for $l-1$, and the proof of the induction step is complete. Therefore, by the principle of induction, Eq.(65) holds for all $l \in [L]$.

Now, let's consider $\|\mathbf{b}^{(l)}\|_\infty$. By Lemma G.6 and Lemma G.5, with probability at least $1 - me^{-c_b^{(l)}\ln^2(m)}$,

$$
\begin{aligned}
\|\mathbf{b}^{(l)}\|_\infty &\leq \|\mathbf{b}_0^{(l)}\|_\infty + \|\mathbf{b}^{(l)} - \mathbf{b}_0^{(l)}\|_\infty \\
&\leq \|\mathbf{b}_0^{(l)}\|_\infty + \|\mathbf{b}^{(l)} - \mathbf{b}_0^{(l)}\| = \tilde{O}\left(\frac{1}{\sqrt{m}}\right).
\end{aligned} \tag{71}
$$

Using union bound, for all $l \in [L]$, we have with probability $1 - m \sum_{l=1}^{L} e^{-c_b^{(l)} \ln^2(m)}$,

$$\|\mathbf{b}^{(l)}\|_\infty = \|\frac{\partial f}{\partial \alpha^{(l)}}\|_\infty = \tilde{O}\left(\frac{1}{\sqrt{m}}\right). \tag{72}$$

$\square$

## H  Generalization to other architectures

In this section, we show that, for both convolutional neural networks (CNN) and residual networks (ResNets), the conditions proposed in Theorem 3.1 are satisfied, and hence these networks have small Hessian spectral norms when the network width $m$ is sufficiently large.

### H.1  Convolutional Neural Networks

A convolutional neural network (CNN) is a network of the type in Eq.(7), with each convolutional layer function $\phi_l$ defined as

$$\alpha^{(l)} = \phi_l(\mathbf{W}^{(l)}; \alpha^{(l-1)}) = \sigma\left(\frac{1}{\sqrt{m_l}} \mathbf{W}^{(l)} * \alpha^{(l-1)}\right), \ \forall l \in [L], \tag{73}$$

where $*$ is the convolution operator (see the definition below), and the layer width $m_l = m$ for all $l = 2, 3, \cdots, L$, and $m_1 = d$ with $d$ as the number of channels of the input.

To simplify the notation, we consider a one-dimensional CNN, i.e., a "image" is an 1-D array of "pixels", and one will find that the analysis in this section also applies to higher dimensional CNNs. We also drop the layer indices $l$, wherever there is no ambiguity.

We denote the number of channels for each hidden layer as $m$, the number of pixels in the "image" as $Q$ and the size of each filter as $K$. Furthermore, we use $i, j \in [m]$ as indices of the channels, $q \in [Q]$ as indices of pixels and $k \in [K]$ as indices within the filter. The input $\alpha \in \mathbb{R}^{m \times Q}$ is a matrix, with $m$ rows as channels and $Q$ columns as pixels. The parameters $\mathbf{W} \in \mathbb{R}^{K \times m \times m}$ is a order 3 tensor. The output of the layer function $\phi$ is of size $m \times Q$. In this 1-D CNN case, the convolution operator is defined as

$$(\mathbf{W} * \alpha)_{i,q} = \sum_{k=1}^{K} \sum_{j=1}^{m} W_{k,i,j} \alpha_{j,q+k-\frac{K+1}{2}}. \tag{74}$$

**Reformulation of convolutional layer.**  Now, we reformulate the convolutional layer function in Eq.(73) into a fully-connected-like function. Then, we can use the techniques developed in Section G to prove for the CNN. Specifically, for all $k \in [K]$, define matrices $W^{[k]}$ and $\alpha^{[k]}$ such that each entry $(W^{[k]})_{ij} = W_{k,i,j}$ and $(\alpha^{[k]})_{jq} = \alpha_{j,q+k-\frac{K+1}{2}}$. Then, the convolution operator in Eq.(74) can be rewritten as

$$(\mathbf{W} * \alpha) = \sum_{k=1}^{K} W^{[k]} \alpha^{[k]}. \tag{75}$$

Here in the summation, it is matrix multiplication. Note that, while $W^{[k]}$ are independent from each other for different $k \in [K]$, the inputs $\alpha^{[k]}$ are not independent from each other; instead, they share pixels: $(\alpha^{[k]})_{j,q} = (\alpha^{[k']})_{j,q+k-k'}$, i.e., each $\alpha^{[k]}$ is a pixel-shifted version of $\alpha$ (newly generated pixels after shift is filled with zeros).

Therefore, the convolutional layer can also be written as (for $l > 1$)

$$\phi(\mathbf{W}; \alpha) = \sigma\left(\sum_{k=1}^{K} \frac{1}{\sqrt{m}} W^{[k]} \alpha^{[k]}\right) \triangleq \sigma(\tilde{\alpha}). \tag{76}$$

Here, we can see we will use this expression of convolutional layer function for analysis in this section.

Before proceeding to the proof for CNN, we first point out a few useful facts, as summarized in the following lemmas.

**Lemma H.1.** *Given matrices $A$, $B$ and $C$ such that $A = BC$, we have $\|A\|_F \leq \|B\|\|C\|_F$, where $\|B\|$ is the spectral norm of matrix $B$.*

See the proof in Appendix I.8. The following two lemmas provide bounds on the spectral norm of weights and Frobenius norm of hidden layers. These two lemmas (and the proofs) are analogous to Lemma G.2 and G.3, and we omit the proof.

**Lemma H.2.** *Suppose the parameters are initialized as $(W_0^{[k]})_{i,j} \sim \mathcal{N}(0, 1)$, for all $k \in [K], i, j \in [m]$. Then, with high probability of the random initialization, we have for any $\mathbf{W} \in B(\mathbf{W}_0, R)$ the following holds*

$$\|W^{[k]}\| = O(\sqrt{m}), \ \forall k \in [K]. \tag{77}$$

**Lemma H.3.** *Suppose the parameters are initialized as $(W_0^{[k]})_{i,j} \sim \mathcal{N}(0, 1)$, for all $k \in [K], i, j \in [m]$ and for all layers. Then, with high probability of the random initialization, we have for any $\mathbf{W} \in B(\mathbf{W}_0, R)$ the following holds at all hidden layers*

$$\|\alpha\|_F = O(\sqrt{m}). \tag{78}$$

*And at the input layer,*

$$\|\alpha\|_F = O(1). \tag{79}$$

In the following, we focus on analyzing the layers with $l > 1$. For the case of $l = 1$, we omit the proof, and refer the readers to the discussion in Section G, which also applies here.

**Proof of Lipschitz continuity.** As seen in Section G, it suffices to prove the boundedness of the operator norms: $\|\nabla_{\mathbf{w}} f\|$ and $\|\nabla_{\alpha} f\|$. Note that, in the convolutional layer function, the vector of parameters $\mathbf{w}$ is reshaped to $\mathbf{W} \in \mathbb{R}^{K \times m \times m}$, and the input is reshaped to $\alpha \in \mathbb{R}^{m \times Q}$. Then, the Euclidean norm of the input becomes Frobenius norm $\|\alpha\|_F$, and the Euclidean norm $\|\mathbf{w}\| = (\sum_{k=1}^{K} \|W^{[k]}\|_F^2)^{1/2}$.

Then, the spectral norm square

$$
\begin{aligned}
\|\nabla_{\alpha}\phi\|^2 &= \frac{1}{m} \sup_{\|V\|_F=1} \sum_{i=1}^{m}\sum_{q=1}^{Q} (\sigma'(\tilde{\alpha}_{i,q}))^2 \Big(\sum_{k=1}^{K} W^{[k]}V^{[k]}\Big)_{i,q}^2 \\
&\leq \frac{1}{m}L_\sigma^2 \sup_{\|V\|_F=1} \Big\|\sum_{k=1}^{K} W^{[k]}V^{[k]}\Big\|_F^2 \\
&\leq \frac{1}{m}L_\sigma^2 \sup_{\|V\|_F=1} \Big(\sum_{k=1}^{K} \|W^{[k]}\|\|V^{[k]}\|_F\Big)^2 \\
&\leq \frac{1}{m}L_\sigma^2 \Big(\sum_{k=1}^{K} \|W^{[k]}\|\Big)^2 \\
&= O(1).
\end{aligned}
$$

Here, in the second inequality, we used Lemma H.1, and in the last equality, we used Lemma H.2. Similarly, using Lemma H.1 and H.3, we also have

$$
\begin{aligned}
\|\nabla_{\mathbf{w}}\phi\|^2 &= \frac{1}{m}\sup\Big\{\sum_{i=1}^{m}\sum_{q=1}^{Q}(\sigma'(\tilde{\alpha}_{i,q}))^2\Big(\sum_{k=1}^{K} V^{[k]}\alpha^{[k]}\Big)_{i,q}^2 : \sum_{k=1}^{K}\|V^{[k]}\|_F^2 = 1\Big\} \\
&\leq \frac{1}{m}L_\sigma^2 \sup\Big\{\Big\|\sum_{k=1}^{K} V^{[k]}\alpha^{[k]}\Big\|_F^2 : \sum_{k=1}^{K}\|V^{[k]}\|_F^2 = 1\Big\} \\
&\leq \frac{1}{m}L_\sigma^2 \sup\Big\{\Big(\sum_{k=1}^{K}\|V^{[k]}\|\|\alpha^{[k]}\|_F\Big)^2 : \sum_{k=1}^{K}\|V^{[k]}\|_F^2 = 1\Big\} \\
&\leq \frac{1}{m}L_\sigma^2 \Big(\sum_{k=1}^{K}\|\alpha^{[k]}\|_F\Big)^2 \\
&= O(1).
\end{aligned}
$$

**Proof of condition (a) in Theorem 3.1.** Now we show that the convolutional layer satisfies the condition (a) proposed in Theorem 3.1. The proof idea is analogous to the fully connected case as in Section G, and we provide a sketch of the proof for CNN here.

Recall that the vector of parameters $\mathbf{w}$ is reshaped to $\mathbf{W} \in \mathbb{R}^{K \times m \times m}$, and the input is reshaped to $\alpha \in \mathbb{R}^{m \times Q}$. Then, by Lemma H.1 and H.2, we have

$$\left\| \frac{\partial^2 \phi}{\partial \alpha^2} \right\|_{2,2,1}$$

$$= \sup \left\{ \sum_{i=1}^{m} \sum_{q=1}^{Q} \frac{1}{m} \left| \sigma''(\tilde{\alpha}_{i,q}) \left( \sum_{k=1}^{K} W^{[k]} V_1^{[k]} \right)_{i,q} \left( \sum_{k=1}^{K} W^{[k]} V_2^{[k]} \right)_{i,q} \right| : \|V_1\|_F = \|V_2\|_F = 1 \right\}$$

$$\leq \frac{\beta_\sigma}{2m} \sup \left\{ \left\| \sum_{k=1}^{K} W^{[k]} V_1^{[k]} \right\|_F^2 + \left\| \sum_{k=1}^{K} W^{[k]} V_2^{[k]} \right\|_F^2 : \|V_1\|_F = \|V_2\|_F = 1 \right\}$$

$$\leq \frac{\beta_\sigma}{2m} \cdot 2 \left( \sum_{k=1}^{K} \|W^{[k]}\| \right)^2$$

$$= O(1).$$

Similarly, by using Lemma H.1, H.2 and H.3, we also have

$$\left\| \frac{\partial^2 \phi}{\partial \alpha \partial \mathbf{w}} \right\|_{2,2,1}$$

$$= \sup \left\{ \sum_{i=1}^{m} \sum_{q=1}^{Q} \frac{1}{m} \left| \sigma''(\tilde{\alpha}_{i,q}) \left( \sum_{k=1}^{K} V_1^{[k]} \alpha^{[k]} \right)_{i,q} \left( \sum_{k=1}^{K} W^{[k]} V_2^{[k]} \right)_{i,q} \right| : \sum_{k=1}^{K} \|V_1^{[k]}\|_F^2 = \|V_2\|_F^2 = 1 \right\}$$

$$\leq \frac{\beta_\sigma}{2m} \sup \left\{ \left\| \sum_{k=1}^{K} V_1^{[k]} \alpha^{[k]} \right\|_F^2 + \left\| \sum_{k=1}^{K} W^{[k]} V_2^{[k]} \right\|_F^2 : \sum_{k=1}^{K} \|V_1^{[k]}\|_F^2 = \|V_2\|_F^2 = 1 \right\}$$

$$\leq \frac{\beta_\sigma}{2m} \cdot \left( \left( \sum_{k=1}^{K} \|\alpha^{[k]}\|_F \right)^2 + \left( \sum_{k=1}^{K} \|W^{[k]}\| \right)^2 \right)$$

$$= O(1),$$

and

$$\left\| \frac{\partial^2 \phi}{\partial \mathbf{w}^2} \right\|_{2,2,1}$$

$$= \sup \left\{ \sum_{i=1}^{m} \sum_{q=1}^{Q} \frac{1}{m} \left| \sigma''(\tilde{\alpha}_{i,q}) \left( \sum_{k=1}^{K} V_1^{[k]} \alpha^{[k]} \right)_{i,q} \left( \sum_{k=1}^{K} V_2^{[k]} \alpha^{[k]} \right)_{i,q} \right| : \sum_{k=1}^{K} \|V_1^{[k]}\|_F^2 = \sum_{k=1}^{K} \|V_2^{[k]}\|_F^2 = 1 \right\}$$

$$\leq \frac{\beta_\sigma}{2m} \sup \left\{ \left\| \sum_{k=1}^{K} V_1^{[k]} \alpha^{[k]} \right\|_F^2 + \left\| \sum_{k=1}^{K} V_2^{[k]} \alpha^{[k]} \right\|_F^2 : \sum_{k=1}^{K} \|V_1^{[k]}\|_F^2 = \sum_{k=1}^{K} \|V_2^{[k]}\|_F^2 = 1 \right\}$$

$$\leq \frac{\beta_\sigma}{2m} \cdot 2 \left( \sum_{k=1}^{K} \|\alpha^{[k]}\|_F \right)^2$$

$$= O(1).$$

**Proof sketch of condition (b) in Theorem 3.1.** The proof idea is similar to the case of fully connected case, as in Section G.3, i.e., proving by induction. The base case of the induction is the same as fully connected case, and we omit it here. The inductive hypothesis for CNN is:
$\max_{i \in [m], q \in [Q]} (\nabla_{\alpha^{(l+1)}} f)_{i,q} = \tilde{O}(1)$.

Now, for $l$-th layer, we have

$$(\nabla_{\alpha^{(l)}} f)_{i,q} = \sum_{j=1}^{m} \sum_{q'=1}^{Q} \sum_{k=1}^{K} (\nabla_{\alpha^{(l+1)}} f)_{j,q'} \sigma'(\tilde{\alpha}_{j,q'}^{(l+1)}) \frac{1}{\sqrt{m}} W_{ji}^{[k]} \mathbb{I}_{q=q'-k+\frac{K+1}{2}}$$

$$= \sum_{k=1}^{K} \sum_{j=1}^{m} (\nabla_{\alpha^{(l+1)}} f)_{j,q+k-\frac{K+1}{2}} \sigma'\left(\tilde{\alpha}_{j,q+k-\frac{K+1}{2}}^{(l+1)}\right) \frac{1}{\sqrt{m}} W_{ji}^{[k]}$$

$$\triangleq \sum_{k=1}^{K} (\nabla_{\alpha^{(l)}} f)_{i,q}^{[k]}.$$

By the same argument as in Section G.3, we have: $\max_{i \in [m], q \in [Q]} (\nabla_{\alpha^{(l)}} f)_{i,q}^{[k]} = \tilde{O}(1)$, for each $k \in [K]$, with high probability of the random initialization. Since $K$ is finite, then we have $\max_{i \in [m], q \in [Q]} (\nabla_{\alpha^{(l)}} f)_{i,q} = \tilde{O}(1)$ with high probability of the random initialization.

## H.2 Residual Networks (ResNet)

In this subsection we prove that the Hessian spectral norm for ResNet also scales as $\tilde{O}(1/\sqrt{m})$, with $m$ being the width of the network. We define the ResNet $f$ as follows:

$$\alpha^{(1)} = \sigma(\frac{1}{\sqrt{d}} W^{(1)} \mathbf{x}),$$

$$\alpha^{(l)} = \sigma(\tilde{\alpha}_{res}^{(l)}) + \alpha^{(l-1)}, \ \tilde{\alpha}_{res}^{(l)} = \frac{1}{\sqrt{m}} W^{(l)} \alpha^{(l-1)}, \ \forall 2 \leq l \leq L,$$

$$f = \frac{1}{\sqrt{m}} \mathbf{v}^T \alpha^{(L)}. \tag{80}$$

The parameters $\mathbf{W} := \{W^{(1)}, W^{(2)}, \cdots, W^{(L)}, W^{(L+1)} := \mathbf{v}\}$ are initialized following the random Gaussian initialization strategy, i.e., $(W_0^{(l)})_{ij} \sim \mathcal{N}(0,1), \forall l \in [L]$, and $v_{0,i} \sim \mathcal{N}(0,1)$, $i, j \in [m]$.

**Remark H.1.** This definition of ResNet differs from the standard ResNet architecture in [He et al.][7] that the skip connections are at every layer, instead of every two layers. One will find that the same analysis can be easily generalized to cases where skip connections are at every two or more layer. The same definition, up to a scaling factor, was also theoretically studied in [7].

We see that the ResNet is the same as a fully connected neural network, Eq. (46), except that the activations $\alpha^{(l)}$ has an extra additive term $\alpha^{(l-1)}$ from the previous layer, interpreted as skip connection. Because of this similarity, the proof for ResNet is almost identical to that for fully connected networks in Section G. In the following, we sketch the proof for ResNet. Specifically, we focus on the arguments that are new to ResNet, and omit those identical to the fully connected case. Hence, we suggest the readers read Section G first.

Parallel to Lemma G.2 and G.3 for fully connected case, we have the following lemmas for the ResNet.

**Lemma H.4.** *Suppose the parameters are initialized as $(W_0^{(l)})_{i,j} \sim \mathcal{N}(0,1)$, for all $l \in [L]$, and $v_{0,i} \sim \mathcal{N}(0,1)$, $i, j \in [m]$. Then, with high probability of the random initialization, we have for any $\mathbf{W} \in B(\mathbf{W}_0, R)$ the following holds*

$$\|W^{(l)}\| = O(\sqrt{m}), \ \forall l \in [L+1]. \tag{81}$$

**Lemma H.5.** *Suppose the parameters are initialized as $(W_0^{(l)})_{i,j} \sim \mathcal{N}(0,1)$, for all $l \in [L]$, and $v_{0,i} \sim \mathcal{N}(0,1)$, $i, j \in [m]$. Then, with high probability of the random initialization, we have for any $\mathbf{W} \in B(\mathbf{W}_0, R)$ the following holds at all hidden layers*

$$\|\alpha^{(l)}\| = O(\sqrt{m}). \tag{82}$$

*Particularly, for the input layer*

$$\|\alpha^{(0)}\| = \|\mathbf{x}\| = O(1). \tag{83}$$

The proofs of the above two lemmas are almost identical to those of Lemma G.2 and G.3. We omit the proofs here, and refer interested readers to proofs of Lemma G.2 and G.3.

**Proofs of Lipschitz continuity for ResNet.** We prove that the first derivatives have bounded spectral norms in the ball $B(\mathbf{W}_0, R)$.

**When** $2 \leq l \leq L$, from the definition of ResNet, Eq.(80), a ResNet layer $\phi_l$ is defined by:

$$\alpha^{(l)} = \phi_l(W^{(l)}; \alpha^{(l-1)}) = \sigma\left(\frac{1}{\sqrt{m}} W^{(l)} \alpha^{(l-1)}\right) + \alpha^{(l-1)}. \tag{84}$$

Therefore, we have

$$\|\nabla_\alpha \phi_l\| = \sup_{\|\mathbf{v}\|=1} \|(\frac{1}{\sqrt{m}} \Sigma'^{(l)} W^{(l)} + I)\mathbf{v}\|$$

$$\leq \sup_{\|\mathbf{v}\|=1} (\frac{1}{\sqrt{m}} \|\Sigma'^{(l)}\| \|W^{(l)}\| \|\mathbf{v}\| + \|\mathbf{v}\|)$$

$$\leq L_\sigma(c_0 + R/\sqrt{m}) + 1 = O(1).$$

We note that $\|\nabla_\mathbf{w} \phi_l\|$ has the same expression as the one of the fully connected networks. By the same argument in Section G.1, as well as Lemma H.5, we have $\|\nabla_\mathbf{w} \phi_l\| = O(1)$.

**When** $l = 1$, the layer function is defined by

$$\alpha^{(1)} = \phi_1(W^{(1)}; \alpha^{(0)}) = \sigma\left(\frac{1}{\sqrt{d}} W^{(1)} \mathbf{x}\right).$$

In this layer, the input $\mathbf{x}$ is fixed (independent of trainable parameters) and not a dynamical variable. Hence, $\nabla_\alpha \phi_1$ is not an interesting object in our Hessian analysis.

And we have

$$\|\nabla_\mathbf{w} \phi_1\| \leq \frac{1}{\sqrt{d}} \|\Sigma'^{(l)}\| \|\mathbf{x}\| \leq L_\sigma C_\mathbf{x} = O(1).$$

We see that both $\|\nabla_\alpha \phi_l\|$ and $\|\nabla_\mathbf{w} \phi_l\|$ are bounded, hence, the (vector valued) layer function of ResNet is Lipschitz continuous.

**Proof of condition (a) in Theorem 3.1 for ResNet.** Note that the skip connection term $\alpha^{(l-1)}$ in Eq.(84) is linear in $\alpha^{(l-1)}$ and independent from $W^{(l)}$. Hence, the order 3 tensors are exactly the same as in the case of fully connected networks. Applying the same argument as in Section G.2 gives the following:

$$\left\|\frac{\partial^2 \phi_l}{(\partial \alpha^{(l-1)})^2}\right\|_{2,2,1} = O(1), \quad \left\|\frac{\partial^2 \phi_l}{\partial \alpha^{(l-1)} \partial W^{(l)}}\right\|_{2,2,1} = O(1), \quad \left\|\frac{\partial^2 \phi_l}{(\partial W^{(l)})^2}\right\|_{2,2,1} = O(1). \tag{85}$$

**Proof of condition (b) in Theorem 3.1 for ResNet.** For a ResNet, define vector $\mathbf{b}_{res}^{(l)} := \nabla_{\alpha^{(l)}} f$ for $l \in [L]$. Specifically, $\mathbf{b}_{res}^{(l)}$ takes the following form:

$$\mathbf{b}_{res}^{(l)} = \prod_{l'=l+1}^{L} \left(\frac{1}{\sqrt{m}} (W^{(l')})^T \Sigma'^{(l')} + I\right) \frac{1}{\sqrt{m}} \mathbf{v}. \tag{86}$$

Compared to the expression of $\mathbf{b}^{(l)}$, in Eq.(62), which is the fully connected network case, the only difference is that $\mathbf{b}_{res}^{(l)}$ for ResNet has an extra additive identity matrix. We argue that the $\infty$-norm $\|\mathbf{b}_{res}^{(l)}\|_\infty$ is still the order of $\tilde{O}(1/\sqrt{m})$. We show this by induction.

First, recall that by the analysis in Section G.3 we have $\|\mathbf{b}^{(l)}\|_\infty = \tilde{O}(\frac{1}{\sqrt{m}})$ for all $l \in [L]$.

In the base case, $\mathbf{b}_{res}^{(L)} = \frac{1}{\sqrt{m}} \mathbf{v} = \mathbf{b}^{(L)}$. Then $\|\mathbf{b}_{res}^{(L)}\|_\infty = \tilde{O}(\frac{1}{\sqrt{m}})$ holds.

Now, suppose $\|\mathbf{b}_{res}^{(l+1)}\|_\infty = \tilde{O}(\frac{1}{\sqrt{m}})$ holds. For $\mathbf{b}_{res}^{(l)}$, we have

$$\mathbf{b}_{res}^{(l)} = \left(\frac{1}{\sqrt{m}}(W^{(l+1)})^T \Sigma'^{(l+1)} + I\right)\mathbf{b}_{res}^{(l+1)} = \frac{1}{\sqrt{m}}(W^{(l+1)})^T \Sigma'^{(l+1)}\mathbf{b}_{res}^{(l+1)} + \mathbf{b}_{res}^{(l+1)}. \quad (87)$$

By an analogous analysis as in Section G.3 and I.7, we have that $\infty$-norm of the first term is of the order $\tilde{O}(1/\sqrt{m})$. Since $\|\mathbf{b}_{res}^{(l+1)}\|_\infty$ is also of the order $\tilde{O}(1/\sqrt{m})$, we conclude that $\|\mathbf{b}_{res}^{(l)}\|_\infty$ is of the order $\tilde{O}(1/\sqrt{m})$.

### H.3  Architecture with mixed layer types

So far, we have seen that fully connected networks (Theorem 3.2), CNNs(Section H.1) and ResNets (Section H.2) satisfy the conditions in Theorem 3.1. In fact, our analysis generalizes to architectures with mixed layer types. Note that the Liptshitz continuity of layer functions and condition (a) of Theorem 3.1 are purely layer-wise, and does not depend on the types of other layers. As for the condition (b), our analysis is inductive: $\|\nabla_{\alpha^{(l)}} f\|_\infty = \tilde{O}(1)$ only relies on the structure of the current layer and the fact that $\|\nabla_{\alpha^{(l+1)}} f\|_\infty = \tilde{O}(1)$.

## I  Proof of Technical Lemmas

### I.1  Proof of Lemma F.1

*Proof.*

$$\|H\| = \left\| \begin{pmatrix} H^{(1,1)} & 0 & \cdots & 0 \\ 0 & 0 & \cdots & 0 \\ \vdots & \vdots & \ddots & \vdots \\ 0 & 0 & \cdots & 0 \end{pmatrix} + \begin{pmatrix} 0 & H^{(1,2)} & \cdots & 0 \\ 0 & 0 & \cdots & 0 \\ \vdots & \vdots & \ddots & \vdots \\ 0 & 0 & \cdots & 0 \end{pmatrix} + \cdots + \begin{pmatrix} 0 & 0 & \cdots & 0 \\ 0 & 0 & \cdots & 0 \\ \vdots & \vdots & \ddots & \vdots \\ 0 & 0 & \cdots & H^{(L+1,L+1)} \end{pmatrix} \right\|$$

$$\leq \sum_{l_1,l_2} \|H^{(l_1,l_2)}\|.$$

$\square$

### I.2  Proofs for Gaussian Random Initialization

*Proof of Lemma G.1.* Consider an arbitrary random matrix $W \in \mathbb{R}^{m_1 \times m_2}$ with each entry $W_{ij} \sim \mathcal{N}(0,1)$. By Corollary 5.35 of [18], for any $t > 0$, we have with probability at least $1 - 2\exp(-\frac{t^2}{2})$,

$$\|W\|_2 \leq \sqrt{m_1} + \sqrt{m_2} + t. \quad (88)$$

In particular, for the initial parameter setting $\mathbf{W}_0$, we have

$$\|W_0^{(1)}\|_2 \leq \sqrt{d} + \sqrt{m} + t,$$
$$\|W_0^{(l)}\|_2 \leq 2\sqrt{m} + t, \quad l \in \{2, 3, ..., L\},$$
$$\|W_0^{(L+1)}\|_2 \leq \sqrt{m} + 1 + t.$$

Letting $t = \sqrt{m}$ and noting that $m > d$, we finish the proof. $\square$

### I.3  Proof of Lemma G.2

*Proof.* By triangle inequality and the definition $\|\mathbf{W}\| = \sum_{l=1}^{L+1}\|W^{(l)}\|_F$, we have for all layers, i.e., $l \in [L+1]$,

$$\|W^{(l)}\|_2 \leq \|W_0^{(l)}\|_2 + \|W^{(l)} - W_0^{(l)}\|_2 \leq \|W_0^{(l)}\|_2 + \|W^{(l)} - W_0^{(l)}\|_F \leq c_0\sqrt{m} + R. \quad (89)$$

Note that, at the output layer, $W^{(L+1)}$ i.e. $\mathbf{v}$ is a vector, and the Frobenius norm $\|\cdot\|_F$ reduces to the Euclidean norm $\|\cdot\|$. $\square$

## I.4  Proof of Lemma G.3

*Proof.* To analyze $\|\alpha^{(l)}(\mathbf{W})\|$, let's first consider the input layer, i.e., $l = 0$: $\|\alpha^{(0)}\| = \|\mathbf{x}\| \leq \sqrt{d}\|\mathbf{x}\|_\infty \leq \sqrt{d}C_\mathbf{x}$, where $d$ is the dimension of the input $\mathbf{x}$. Then we prove Eq.(49) by induction. For the first hidden layer $l = 1$,

$$
\begin{aligned}
\|\alpha^{(1)}(\mathbf{W})\| &= \left\| \sigma\left( \frac{1}{\sqrt{d}} W^{(1)}\alpha^{(0)} \right) \right\| \\
&\leq \frac{1}{\sqrt{d}} L_\sigma \|W^{(1)}\|_2 \|\alpha^{(0)}\| + \sigma(0) \\
&\leq \frac{1}{\sqrt{d}} L_\sigma (c_0\sqrt{m} + R)\|\alpha^{(0)}\| + \sigma(0) \\
&\leq L_\sigma (c_0 + R/\sqrt{m})\sqrt{m}C_\mathbf{x} + \sigma(0).
\end{aligned}
\tag{90}
$$

Above, we used the $L_\sigma$-Lipschitz continuity and applied Lemma G.2 in the second inequality. Now, suppose for $l$-th layer we have

$$
\|\alpha^{(l)}(\mathbf{W})\| \leq L_\sigma^l (c_0 + R/\sqrt{m})^l \sqrt{m}C_\mathbf{x} + \sum_{i=1}^{l} L_\sigma^{i-1}(c_0 + R/\sqrt{m})^{i-1}\sigma(0).
\tag{91}
$$

Then, by a similar argument as in Eq.(90), we can get

$$
\|\alpha^{(l+1)}(\mathbf{W})\| = \left\| \sigma\left( \frac{1}{\sqrt{m}} W^{(l+1)}\alpha^{(l)}(\mathbf{W}) \right) \right\|
$$

$$
\leq L_\sigma^{l+1}(c_0 + R/\sqrt{m})^{l+1}\sqrt{m}C_\mathbf{x} + \sum_{i=1}^{l+1} L_\sigma^{i-1}(c_0 + R/\sqrt{m})^{i-1}\sigma(0).
$$

$\square$

## I.5  Proof of Lemma G.4

*Proof.* When $2 \leq l \leq L$, $|\alpha_i^{(l)}|$ takes the following form:

$$
\begin{aligned}
|\alpha_i^{(l)}| &= \left| \sigma\left( \frac{1}{\sqrt{m}} \sum_{k=1}^{m} W_{ik}^{(l)}\alpha_k^{(l-1)} \right) \right| \\
&\leq \left| \frac{L_\sigma}{\sqrt{m}} \sum_{k=1}^{m} W_{ik}^{(l)}\alpha_k^{(l-1)} \right| + |\sigma(0)|,
\end{aligned}
$$

where we can see $\sum_{k=1}^{m} W_{ik}^{(l)}\alpha_k^{(l-1)} \sim \mathcal{N}(0, \|\alpha^{(l-1)}\|^2)$ since $W_{ik}^{(l)} \sim \mathcal{N}(0, 1)$ at initialization. By the concentration inequality for Gaussian random variable, we have

$$
\begin{aligned}
\mathbb{P}[|\alpha_i^{(l)}| \geq \ln(m) + |\sigma(0)|] &\leq \mathbb{P}\left[ \left| \frac{L_\sigma}{\sqrt{m}} \sum_{k=1}^{m} W_{ik}^{(l)}\alpha_k^{(l-1)} \right| \geq \ln(m) \right] \\
&\leq 2e^{-\frac{m\ln^2(m)}{2L_\sigma^2 \|\alpha^{(l-1)}\|^2}} \\
&= 2e^{-c_\alpha^{(l)}\ln^2(m)},
\end{aligned}
$$

for $c_\alpha^{(l)} = \frac{m}{2L_\sigma^2 \|\alpha^{(l-1)}\|^2} = \Omega(1)$ by Lemma G.3.
When $l = 1$, we have

$$
\begin{aligned}
|\alpha_i^{(1)}| &= \left| \sigma\left( \frac{1}{\sqrt{d}} \sum_{k=1}^{d} W_{ik}^{(1)}\mathbf{x}_k \right) \right| \\
&\leq \left| \frac{L_\sigma}{\sqrt{d}} \sum_{k=1}^{d} W_{ik}^{(1)}\mathbf{x}_k \right| + |\sigma(0)|.
\end{aligned}
$$

Similarly, at initialization, $\sum_{k=1}^{d} W_{ik}^{(1)} \mathbf{x}_k \sim \mathcal{N}(0, \|\mathbf{x}\|^2)$. Hence

$$\mathbb{P}[|\alpha_i^{(1)}| \geq \ln(m) + |\sigma(0)|] \leq \mathbb{P}\left[\left|\frac{L_\sigma}{\sqrt{d}} \sum_{k=1}^{d} W_{ik}^{(l)} \mathbf{x}_k\right| \geq \ln(m)\right]$$

$$\leq 2e^{-\frac{d\ln^2(m)}{2L_\sigma^2\|\mathbf{x}\|^2}}$$

$$= 2e^{-c_\alpha^{(0)}\ln^2(m)},$$

where we denote $\frac{d}{L_\sigma^2\|\mathbf{x}\|}$ by $c_\alpha^{(0)}$, which is of the order $\Theta(1)$.

Therefore, $|\alpha_i^{(l)}| = \tilde{O}(1)$ with probability at least $1 - 2e^{-c_\alpha^{(l)}\ln^2(m)}$ for all $l \in [L]$. $\qquad\square$

## I.6 Proof of Lemma G.5

*Proof.* The expression of the derivatives $\mathbf{b}^{(l)}$ is

$$\mathbf{b}^{(l)} = \left(\prod_{l'=l+1}^{L} \frac{1}{\sqrt{m}} (W^{(l')})^T \Sigma'^{(l')}\right) \frac{1}{\sqrt{m}} \mathbf{v}, \tag{92}$$

where $\Sigma'^{(l')}$ is a diagonal matrix with each non-zero element $(\Sigma'^{(l')})_{ii} = \sigma'(\tilde{\alpha}_i^{(l)}(\mathbf{W}))$.
We prove the lemma by induction. When $l = L$, using Lemma G.2, we have

$$\|\mathbf{b}^{(L)}\| = \frac{1}{\sqrt{m}} \|\mathbf{v}\| \leq \frac{1}{\sqrt{m}} (c_0 \sqrt{m} + R) = c_0 + R/\sqrt{m}. \tag{93}$$

Suppose at $l$-th layer, $\|\mathbf{b}^{(l)}\| \leq L_\sigma^{L-l}(c_0 + R/\sqrt{m})^{L-l+1}$. Then

$$\|\mathbf{b}^{(l-1)}\| = \|\frac{1}{\sqrt{m}} (W^{(l)})^T \Sigma'^{(l)} \mathbf{b}^{(l)}\|$$

$$\leq \frac{1}{\sqrt{m}} \|W^{(l)}\|_2 \|\Sigma'^{(l)}\|_2 \|\mathbf{b}^{(l)}\|$$

$$\leq (c_0 + R/\sqrt{m}) L_\sigma \|\mathbf{b}^{(l)}\|$$

$$\leq L_\sigma^{L-l+1}(c_0 + R/\sqrt{m})^{L-l+2}.$$

Above, we used Lemma G.2 and the $L_\sigma$-Lipschitz continuity of the activation function $\sigma(\cdot)$ in the second inequality.
Setting $R = 0$, we immediately obtain Eq.(64).

$\qquad\square$

## I.7 Proof of Lemma G.6

*Proof.* We prove it by induction. When $l = L$, $\mathbf{b}_0^{(L)} = \frac{1}{\sqrt{m}} \mathbf{v}_0$. Since $\mathbf{v}_{0,i} \sim \mathcal{N}(0, 1)$, by the concentration inequality, for every $i \in [m]$, we have

$$\mathbb{P}[|\mathbf{v}_{0,i}| \geq \ln(m)] \leq 2e^{\frac{-\ln^2(m)}{2}}.$$

By union bound, with probability at least $1 - 2me^{\frac{-\ln^2(m)}{2}}$,

$$\|\mathbf{v}_0\|_\infty \leq \ln(m),$$

in other words,

$$\|\mathbf{b}_0^{(L)}\|_\infty = \tilde{O}(1/\sqrt{m}).$$

Supposing with probability $1 - me^{-c_b^{(l)}\ln^2(m)}$ for some constant $c_b^{(l)} > 0$, we have $\|\mathbf{b}_0^{(l)}\|_\infty = \tilde{O}(1/\sqrt{m})$. Then we show $\|\mathbf{b}_0^{(l-1)}\|_\infty = \tilde{O}(1/\sqrt{m})$ with probability $1 - me^{-c_b^{(l-1)}\ln^2(m)}$ for some constant $c_b^{(l-1)} > 0$.

For simplicity, in the rest of the proof, we hide the subscript 0. Hence we denote $\mathbf{b}_0^{(l-1)} = \frac{1}{\sqrt{m}}(W_0^{(l-1)})^T \Sigma_0'^{(l-1)} \mathbf{b}_0^{(l)}$ by

$$\mathbf{b}^{(l-1)} = \frac{1}{\sqrt{m}}(W^{(l-1)})^T \Sigma'^{(l-1)} \mathbf{b}^{(l)},$$

where $(W^{(l-1)})_{ij} \sim \mathcal{N}(0,1)$.

Similarly, we analyze every component of $\mathbf{b}^{(l-1)}$:

$$|\mathbf{b}_i^{(l-1)}| = \left| \frac{1}{\sqrt{m}} \sum_{k=1}^{m} W_{ki}^{(l-1)} \sigma' \left( \frac{1}{\sqrt{m}} \sum_{j=1}^{m} W_{kj}^{(l-1)} \alpha_j^{(l-2)} \right) \mathbf{b}_k^{(l)} \right|$$

$$\leq \left| \frac{1}{\sqrt{m}} \sum_{k=1}^{m} W_{ki}^{(l-1)} \sigma' \left( \frac{1}{\sqrt{m}} \sum_{j \neq i}^{m} W_{kj}^{(l-1)} \alpha_j^{(l-2)} \right) \mathbf{b}_k^{(l)} + \frac{1}{m} \beta_\sigma \alpha_i^{(l-2)} \sum_{k=1}^{m} (W_{ki}^{(l-1)})^2 \mathbf{b}_k^{(l)} \right|$$

$$\leq \left| \frac{1}{\sqrt{m}} \sum_{k=1}^{m} W_{ki}^{(l-1)} \sigma' \left( \frac{1}{\sqrt{m}} \sum_{j \neq i}^{m} W_{kj}^{(l-1)} \alpha_j^{(l-2)} \right) \mathbf{b}_k^{(l)} \right| + \left| \frac{1}{m} \beta_\sigma \alpha_i^{(l-2)} \sum_{k=1}^{m} (W_{ki}^{(l-1)})^2 \mathbf{b}_k^{(l)} \right|.$$

For the first term, we use a Gaussian random variable to bound it:

$$\frac{1}{\sqrt{m}} \sum_{k=1}^{m} W_{ki}^{(l-1)} \sigma' \left( \frac{1}{\sqrt{m}} \sum_{j \neq i}^{m} W_{kj}^{(l-1)} \alpha_j^{(l-2)} \right) \mathbf{b}_k^{(l)} \leq \frac{L_\sigma}{\sqrt{m}} \sum_{k=1}^{m} W_{ki}^{(l-1)} \mathbf{b}_k^{(l)} \sim \mathcal{N} \left( 0, \frac{L_\sigma^2}{m} \|\mathbf{b}^{(l)}\|^2 \right).$$

Using the concentration inequality, we have

$$\mathbb{P} \left[ \left| \frac{L_\sigma}{\sqrt{m}} \sum_{k=1}^{m} W_{ki}^{(l-1)} \mathbf{b}_k^{(l)} \right| \geq \frac{\ln(m)}{\sqrt{m}} \right] \leq 2e^{-\frac{\ln^2(m)}{2L_\sigma^2 \|\mathbf{b}^{(l)}\|^2}} \leq 2e^{-c_\sigma^{(l)} \ln^2(m)},$$

for some $c_\sigma^{(l)} = \frac{1}{2L_\sigma^2 \|\mathbf{b}^{(l)}\|^2} \geq \frac{1}{2L_\sigma^{2L-2l+2} c_0^{2L-2l+2}}$ by Lemma G.5.

For the second term, we have

$$\frac{1}{m} \beta_\sigma \alpha_i^{(l-2)} \sum_{k=1}^{m} (W_{ki}^{(l-1)})^2 \mathbf{b}_k^{(l)} \leq \frac{1}{m} \beta_\sigma |\alpha_i^{(l-2)}| \|\mathbf{b}^{(l)}\|_\infty \sum_{k=1}^{m} (W_{ki}^{(l-1)})^2,$$

where we can see $\sum_{k=1}^{m} (W_{ki}^{(l-1)})^2 \sim \chi^2(m)$.

By Lemma G.4, with probability $1 - e^{-c_\alpha^{(l-2)} \ln^2(m)}$, we get $|\alpha_i^{(l-2)}| = \tilde{O}(1)$. Hence, by Lemma 1 in [12], there exist constants $\tilde{c}_1, \tilde{c}_2, \tilde{c}_3 > 0$, such that

$$\mathbb{P} \left[ \frac{1}{m} \beta_\sigma |\alpha_i^{(l-2)}| \|\mathbf{b}^{(l)}\|_\infty \sum_{k=1}^{m} (W_{ki}^{(l-1)})^2 \geq \tilde{c}_1 \frac{\ln^{\tilde{c}_3}(m)}{\sqrt{m}} \right] \leq e^{-\tilde{c}_2 m},$$

with probability $1 - me^{-c_b^{(l)} \ln^2(m)}$ by the induction hypothesis.

Combining these probability terms, there exists a constant $c_b^{(l-1)}$ such that

$$e^{-c_b^{(l-1)} \ln^2(m)} \leq me^{-c_b^{(l)} \ln^2(m)} + 2e^{-c_\sigma^{(l)} \ln^2(m)} + 2e^{-c_\alpha^{(l-2)} \ln^2(m)} + e^{-\tilde{c}_2 m}.$$

Then with probability at least $1 - e^{-c_b^{(l-1)} \ln^2(m)}$,

$$|\mathbf{b}_i^{(l-1)}| = \tilde{O}(1/\sqrt{m}).$$

By union bound, with probability at least $1 - me^{-c_b^{(l-1)} \ln^2(m)}$, we have

$$\|\mathbf{b}^{(l-1)}\|_\infty = \tilde{O}(1/\sqrt{m}).$$

Hence by the principle of induction, for all $l \in [L]$, with probability at least $1 - me^{-c_b^{(l)} \ln^2(m)}$ for some constant $c_b^{(l)} > 0$, we have

$$\|\mathbf{b}^{(l)}\|_\infty = \tilde{O}(1/\sqrt{m}). \tag{94}$$

$\square$

## I.8 Proof of Lemma H.1

*Proof.* Let $A = (\mathbf{a}_1, \mathbf{a}_2, \cdots, \mathbf{a}_d)$ and $C = (\mathbf{c}_1, \mathbf{c}_2, \cdots, \mathbf{c}_d)$, where each $\mathbf{a}_i$ is a column of the matrix $A$ and each $\mathbf{c}_i$ is a column of the matrix $C$. Then we have

$$\mathbf{a}_i = B\mathbf{c}_i, \ \forall i \in [d]. \tag{95}$$

Now, for the Frobenius norm, we have

$$\|A\|_F^2 = \sum_{i=1}^d \|\mathbf{a}_i\|^2 = \sum_{i=1}^d \|B\mathbf{c}_i\|^2 \leq \sum_{i=1}^d \|B\|^2 \|\mathbf{c}_i\|^2 = \|B\|^2 \|C\|_F^2.$$

$\square$

Hence, $\|A\|_F \leq \|B\| \|C\|_F$.

## Footnotes

[5]LeCun, Yann A and Bottou, Léon and Orr, Genevieve B and Müller, Klaus-Robert,"Efficient backprop". In:*Neural networks: Tricks of the trade.* Springer, 2012, pp. 9-48.

[6] Indeed, it does not show up in the Hessian analysis, see Section F.

[7]Kaiming He, Xiangyu Zhang, Shaoqing Ren, and Jian Sun. "Deep residual learning for image recognition". In: *Proceedings of the IEEE conference on computer vision and pattern recognition.* 2016, pp. 770-778.