[Reviews · NeurIPS 2020]

Review 1

Summary and Contributions: The paper gives a general condition to make NN have the constant tangent kernel, i.e., the training of the NN can be described as a training of NTK. It is a generalization/summarization of many previous NTK papers.

Strengths: - The theory in this paper is sound, nicely written and easy to understand. I believe the proofs are correct. - The experiments in the paper are convincing. - It is definitely relevant to the NeurIPS community.

Weaknesses: - Despite the fact the theory is sound, the authors didn't acknowledge previous works properly. For example, Theorem 3.2 and results in Appendix G has been proved previously (e.g., [1]). The paper fails to cite and/or compare some of the previous works. - Given the fact that the applications of Theorem 3.1 are mostly proved previously, the contribution of the paper is not so significant. I tend to think of this paper as a summarization of the previous line of NTK papers. [1] Sanjeev Arora, Simon S. Du, Wei Hu, Zhiyuan Li, Ruslan Salakhutdinov, Ruosong Wang. On Exact Computation with an Infinitely Wide Neural Net.

Correctness: I believe the claims and the experiments are correct in this paper.

Clarity: The paper is well written and easy to understand. Minor: - line 144, it is better to mention x bounded earlier.

Relation to Prior Work: The paper fails to cite many closely related works, not to mention a discussion between its results and previous results, especially in techinical aspects. ---- post author response comments ---- The authors have addressed my main concerns, and I have changed the score accordingly. I still strongly recommend the authors to acknowledge and compare previous paper that used different proofs but have the same conclusion. (e.g., people have proved infinitely wide FC/CNN is kernel, let readers know the differences in proof. )

Reproducibility: Yes

Additional Feedback: I think many NTK works has the spirit of this paper, can you give an application of Theorem 3.1 that haven't been studied yet.


Review 2

Summary and Contributions: The paper sheds light on recent theoretical development of optimization of overparameterized neural networks through the lens of constancy of neural tangent kernel. First, the authors relate the constancy of a tangent kernel to a linear model in weights (Proposition 2.1.). With this, the authors could identify conditions for the constant kernel with respect to the spectral norm of the Hessian. Given this tool, the paper pinpoints a few key assumptions, linearity of output and sparse dependence of activation function, and no-bottleneck for guaranteeing constancy of the tangent kernel.

Strengths: I believe the work provides fundamental insights on recent developments in theory of deep learning pioneered by NTK. Various strong breakthroughs including convergence of (S)GD for overparameterized networks has been developed relying on the constancy of NTK during training. This work sheds light on the conditions when those assumptions could break down and even offer solutions when it could be remedied despite non-constancy of NTK.

Weaknesses: It would be useful to have perspective on how far current theoretical developments are to the actual practical case. For example, the paper mostly focuses on squared loss while widely applied NNs use softmax-cross entropy loss. I would encourage putting discussion on how one should think about optimization of those networks in the context of this paper’s result. Another point to raise is that while the claim of Section 5 and Theorem 5.1 is interesting and powerful, most discussion is based on a separate paper included in supplementary material. It does have significant overlap with current submission, so if provided submission is published in another archival venue or is another submission to NeurIPS it may be subject to dual submission (https://neurips.cc/Conferences/2020/CallForPapers). Otherwise, one should extract relevant parts such as proof of Theorem 5.1, or just cite a separate paper distinguishing contribution.

Correctness: Caveat: I have not followed detailed proof in the appendix but theoretical claims sound firm and consistent with other known works. Also in section 6 claims are reasonably backed well by empirical support.

Clarity: Paper is well written and easy to identify core statements.

Relation to Prior Work: Discussion on recent developments and prior work is clearly discussed and mostly does a good job of stating the contribution related to the prior work. (Caveat: I am more familiar with literature related to NTK but not familiar with general non-convex optimization literature.)

Reproducibility: Yes

Additional Feedback: [Post Author Response] I thank the authors for responding to concerns and questions, which made me appreciate the paper better. As clarified by the authors there won't be issues with dual submission. I think the submission is good submission and will be general interest to NeurIPS community and suggest accepting. As regards to softmax, I agree with the authors when the output is softmax that current paper analysis holds. It would be interesting what would happen with softmax nonlinearities that appears in self-attention layers of Transformer architectures. -------------------------------------------------------- Few comments and nits: 1) Specify or define that Hessian is not of the loss but of the function F. In general, I believe readers at first pass may confuse Hessian to be that of the loss since it is more commonly referred to in deep learning literature. 2) L16-17: NTK is not constant in all w but should specify a domain. If you choose w far from initialization you could find w* where K deviates by a lot 3) L66: is -> as 4) It may be beneficial to provide examples of popular activation functions that satisfy (and do not satisfy) \beta_{\sigma} - smooth conditions used in the assumptions. Questions to the authors: 1) What is the important point of emphasizing Euclidean norm change is O(1)? It is also implicitly discussed in [1] where small parameter change to O(1) function change is discussed. Should I understand the point to be while literature casually talks about “small weight change”, one should be aware that in Euclidean norm it could be O(1) due to large dimensions? 2) There are few non-sparse non-linearity used in deep learning. Few examples I can think of are softmax, maxout. In the context of self-attention layers, softmax is becoming increasingly important. Is the condition (b) in Theorem 3.1 violated for these non-linearities and does not have a constancy guarantee? Do authors believe these networks would not have constant tangent kernels? 3) L251: logically violation of conditions in Theorem 3.1 does not necessarily lead to breakdown of linearity since Theorem 3.1 is not if and only if statement, correct? [1] Lee et al., Wide Neural Networks of Any Depth Evolve as Linear Models Under Gradient Descent, NeurIPS 2019


Review 3

Summary and Contributions: The authors analyze under what conditions the Neural Tangent Kernel(NTK) remains constant, offering an important insight into the theoretical analyses of asymptotically wide neural networks. The crucial insights are twofold, first showing that a Riemannian manifold with metric defined by the NTK has constant gradients if and only if the manifold is linear. Secondly, the authors use elementary results from analysis to show that a vanishing spectral norm of the Hessian tensor inextricably implies linearity of the model. Technically proving the result requires showing the (approximate) sparsity of the Hessian wrt. to preactivations, and chaining the upper bound on the spectral norm across the layers. In conjunction with results from random matrix theory, it allows the authors to show that the spectral norm of a general L layer network without bottlenecks and with linear output is of order O(\sqrt{m}) where m is the width of all the layers. The theoretical result importantly hinges on the linearity of the output layer and the absence of bottleneck layers. Under these conditions, gradient descent is guaranteed to converge to a global minimizer of the loss function. The authors support their theoretical results with numerical experiments showing that a) models satisfying the aforesaid conditions the achieve exponential convergence to a global optimum b) Models with non-linear output and/or with bottleneck layer have do not have a constant kernel during training

Strengths: The submission offers a novel, theoretical sound condition that guarantees the behavior of the NTK limit throughout training. This, as pointed out by the authors, is in stark contrast to the so called ‘lazy training regime’ that assumes infinitesimal changes in weights due to the infinitesimal learning rate (gradient descent vs gradient flow over finite horizon). The results presented in this paper therefore offer an important theoretical contribution to the understanding of ‘infinitely wide’ neural networks. In addition to the theoretical contribution, the work is also of potential practical interest. While NTK have not been widely adopted by machine learning practitioners, this work gives clear conditions under which the randomized kernel defined by a neural network remains constant and therefore might enjoy performance guarantees known for kernel machines; i.e. this includes generalization, but also the superfluousness of using gradient flows in place of standard gradient descent. The empirical evaluation leaves nothing to be desired and offers good support for the authors’ theoretical results.

Weaknesses: None that are apparent. See additional feedback for potential ways of broadening the practical impact of these results.

Correctness: The reviewer carefully read the proofs, and the empirical methodology and both are to the best of the reviewers’ understanding correct and well executed

Clarity: The paper is very clearly written, subjectively striking a balance between rigor and readability. The flow of the submission is impeccable.

Relation to Prior Work: The authors acknowledge all the relevant literature, and their work very extends previous results. As mentioned before it offers very clear, and insightful conditions under which the previously analyze Neural Tangent Kernel remains constant. To the best of the reviewers knowledge that question has remained open since the original publication by Jacot et al. in 2018. The discussion of the differences between Chizat et al and the current work is satisfactory, but perhaps stands to be elaborated more – i.e. while the current submission makes little assumptions about the readers’ familiarity with the relevant material, it could perhaps position itself better and why the difference between the reported results and the `lazy training` regime are significant.

Reproducibility: Yes

Additional Feedback: I would like to raise two suggestions that might broaden the practical relevance of these results: 1) The original derivation by Jacot as well as the results from `lazy training` by Chizat require that the NTK be optimized using gradient flows. It seems that the current results imply that (discrete-time) gradient descent ought to be sufficient, thereby removing the need for integrating a high-dimensional ODE. It seems that this might lower the `entry point` for practitioners and make NTKs are more readily available tool. 2) This suggestion is somewhat conjectural, but it seems that alternative losses for common classification tasks might enjoy additional theoretical benefits. Beyond using the Brier score, could the authors comment on the applicability of their results assuming a hinge loss? It seems that the hinge loss ought to satisfy conditions in Eq(13) and Eq(14), given that its second derivative is zero almost everywhere.

[Author Response · NeurIPS 2020]

We thank all reviewers for the insightful and encouraging comments. Below we provide a point by point response to
Reviewers 1,2,3 (R1, R2, R3).

**R1 + R2:** [Difference from "lazy training"]. One key contribution of our work is that we identify constancy of tangent
kernel, first observed in Jacot et al. (2018), as related to the scaling of the Hessian norm. Notice that the constancy of
the tangent kernel cannot be explained from the point of view of "lazy training": when the last layer is non-linear, the
change of parameter from the initialization is of the same order as for the linear case, but the tangent kernel is no longer
constant along the optimization path, as the Hessian norm is no longer small (see Section 4).

**R1:** *Theorem 3.2 and results in Appendix G has been proved previously (e.g., [1]). ... I tend to think of this paper as a*
*summarization of the previous line of NTK papers. ([1] Sanjeev Arora, et al. On Exact Computation...)*

Our Hessian analysis results, including Theorem 3.2 and Theorem 3.1, are new. Note that previous works, including
[1], only analyze the tangent kernel matrix, which is first order. In contrast, we analyze the Hessian, a second order
differential operator. We note that in some related works, including [1], the notation $H$ stands for the tangent kernel,
while we use it to denote the Hessian matrix. This can perhaps cause confusion. Our novel contributions include
identifying the underlying reasons for constancy of NTK (small Hessian norm, as opposed to "lazy training"), and the
finding that NTK is **not constant** when the last layer is non-linear, even in the infinite width limit (Section 4).

**R2:** *The paper mostly focuses on squared loss while widely applied NNs use softmax-cross entropy loss. I would*
*encourage putting discussion on ... optimization of those networks in the context of this paper's result.*

Thank you for the suggestion. The main focus in this submission is to uncover the underlying reasons for the constancy
of NTK (which depends only on the model, rather than the loss function). Still, it is an important issue, we will add a
discussion.

**R2:** *It (supplementary B) does have significant overlap with current submission ... may be subject to dual submission ...*
*or just cite a separate paper distinguishing contribution.*

There is no dual submission issue as the supplementary B has not been submitted to NeurIPS or any other confer-
ence/journal. Given the space constraints, it does not seem feasible to have a full discussion of the optimization-related
issues. For the final version, we are planning to cite the optimization results as a separate document.

**R2:** *What is the important point of emphasizing Euclidean norm change is O(1)? ... Should I understand the point to be*
*while literature casually talks about "small weight change", one should be aware that in Euclidean norm it could be*
*O(1) due to large dimensions?*

Yes. The measurement of the change of parameters from initialization depends on the norm. In dimension $m$, the
Euclidean norm and infinity-norm can be different by a factor of $\sqrt{m}$. When dimension increases, the infinity norm of
the difference from the initialization to the solution converges to zero. However, the Euclidean norm of the difference is
always $O(1)$. Importantly, the remainder term of the Taylor expansion (and hence the constancy of TK) is controlled by
the Euclidean norm of the difference, not the infinity norm. In contrast, "lazy training" suggests that the optimization
path stays close to the initialization point. We will clarify this in the paper.

**R2:** *Is the condition (b) in Theorem 3.1 violated for these non-linearities (softmax, maxout) and does not have a*
*constancy guarantee? Do authors believe these networks would not have constant tangent kernels*

This is an open question so far. These non-linearities do not fit in our current analysis. If softmax is in the output layer,
our analysis in Section 4 shows that the model does not have a constant tangent kernel.

**R2:** *L251: logically violation of conditions in Theorem 3.1 does not necessarily lead to breakdown of linearity since*
*Theorem 3.1 is not if and only if statement, correct?*

Yes. The conditions in Thm 3.1 are sufficient but not necessary. But note that the neural networks that are shown to
have constant NTK satisfy these conditions. Hence, Thm 3.1 is enough to explain the phenomenon of constant NTK.

**R3:** Thank you for the positive comments and the helpful suggestions.

**R3:** *Could the authors comment on the applicability of their results assuming a hinge loss? It seems that the hinge loss*
*ought to satisfy conditions in Eq(13) and Eq(14), given that its second derivative is zero almost everywhere.*

First, note that the Hessian is defined for the model, not the loss function. For hinge-loss-like activation functions, the
Hessian is zero at most locations, but it is infinite at the hinge point. We note that the Hessian affects the linearity of
the model (i.e., constancy of tangent kernel) through Taylor expansion, see Proposition 2.1 and its proof. Hence, the
tangent kernel, which is the first-derivative, depends on the integral of the Hessian, which is second-derivative. This
infinite Hessian for hinge-like activation functions is likely to have a non-trivial contribution to the tangent kernel after
integration, implying non-constant tangent kernel.

[Meta-Review · NeurIPS 2020]

This paper clarify the condition under which the NTK remains constant. First, it is pointed out that the NTK is constant if and only if the model is linear. Second, it is shown that the NTK is almost constant if the spectral norm of the Hessian is small. The Hessian norm is bounded by some conditions: linearity of output, sparse dependence of activation function, and no-bottleneck layers. Overall, this paper is well written. Clarifying conditions under which constancy of NTK is quite beneficial to wide range of audiences especially who are working on a infinite width network training. On the other hand, several comments are made from reviewers to improve the paper. I encourage the authors to reflect them to the final version as much as possible.